# Learning Bregman Divergences with Application to Robustness

**Mohamed-Hicham Leghettas**
Department of Computer Science
ETH Zurich, Switzerland
mleghettas@inf.ethz.ch

**Markus Püschel**
Department of Computer Science
ETH Zurich, Switzerland
pueschel@inf.ethz.ch

## Abstract

We propose a novel and general method to learn Bregman divergences from raw high-dimensional data that measure similarity between images in pixel space. As a prototypical application, we learn divergences that consider real-world corruptions of images (e.g., blur) as close to the original and noisy perturbations as far, even if in $L^p$-distance the opposite holds. We also show that the learned Bregman divergence excels on datasets of human perceptual similarity judgment, suggesting its utility in a range of applications. We then define adversarial attacks by replacing the projected gradient descent (PGD) with the mirror descent associated with the learned Bregman divergence, and use them to improve the state-of-the-art in robustness through adversarial training for common image corruptions. In particular, for the contrast corruption that was found problematic in prior work we achieve an accuracy that exceeds the $L^p$- and the LPIPS-based adversarially trained neural networks by a margin of 27.16% on the CIFAR-10-C corruption data set.

## 1 Introduction

The need to measure the semantic distance between images is a recurring requirement in various computer vision tasks, including image retrieval [55, 53, 2], near-duplicate detection [82], face recognition [64], and zero-shot learning [60]. This has led to a significant body of research in the field of metric learning [75, 7], which focuses on developing automated methods for learning such distances. The most successful approaches to assessing similarity between images involve encoding them into a compact latent space and computing the $L^p$-distance between the resulting latent features. Image encoders are typically residual neural networks or vision transformers that are pre-trained in a supervised [78], weakly-supervised [36], or self-supervised [31] fashion. The latent space is usually assumed Euclidean and hence the $L^2$-norm is the common choice, although some non-Euclidean geometries have been considered [23].

Image similarity measures are also crucial in the field of robust machine learning. Since models are known to be sensitive to small input perturbations [9, 68, 56], a robustness study requires a measure for the difference between clean and perturbed inputs. A common choice is the $L^p$-norm computed in the pixel space. It lacks semantic meaning but adversarial training (AT) for robustness using these norms (via adversarial training [52] and its many follow-up variants, e.g., [69, 80, 12, 72, 14, 61, 37]) has been found to also improve the robustness to distribution shifts associated with common, realistic image corruptions like blur or contrast changes [20, 33]. Conversely, corruption robustness evaluation is shown more reliable than adversarial robustness evaluations when distinguishing successful adversarial defense methods from ones that merely cause vanishing gradients [25].

Both metric learning and corruption robustness approaches obtain similarity measures by calculating standard norms in latent spaces. In this work, we take a different route by learning Bregman divergences directly in the pixel space. This way, we benefit from a strong mathematical underpinning

38th Conference on Neural Information Processing Systems (NeurIPS 2024).

including the associated *mirror descent*, an optimization framework to natively solve constrained problems that we then put to use for AT.

**Bregman divergence and mirror descent.** The Bregman divergence [10] (referred to as BD in the remaining paper) is a generalization of the Kullback–Leibler (KL) divergence [45], and is widely used in statistics and information theory to define distances in spaces where the Euclidean geometry is not appropriate such as probability distributions, covariance descriptors, random processes and others [16, 18, 6, 67, 27, 29]. It is defined via an underlying base function (e.g., the Shannon entropy for the KL divergence) that has to be strongly convex and with invertible gradient. Nemirovski and Yudin introduced the mirror descent framework [54] as a method for minimizing a function by utilizing a Bregman divergence to incorporate the geometric structure of the underlying space.

**Contributions.** In this paper we offer progress in the quest for similarity measures through a theoretically principled approach to learn BDs for images in pixel space and exploit the associated mirror descent for achieving robustness through AT. Our main contributions are as follows:

- We provide a novel self-supervised algorithm to derive BDs for images in pixel space. The key idea is to learn eligible base functions using a suitable network architecture. These divergences are semantic in the sense that they assess similar images as close and randomly perturbed ones as far from the clean image, even if in Euclidean distance the converse holds.

- We then learn first BDs that are corruption-specific, where similar images are derived by applying image corruptions from CIFAR-10-C dataset [33]. Then we learn BDs that are corruption-oblivious where similar images are obtained from Berkeley-Adobe Perceptual Patch Similarity (BAPPS) dataset [81].

- We show that the learned BDs are consistent and successfully distinguish between corrupted and noisy images. We also show that a BD learned to mimic human judgment on the BAPPS dataset performs well on the two alternative forced choice (2AFC) test.

- We then propose a mirror-descent-inspired algorithm to perform semantic adversarial attacks using the learned BDs instead of the $L^p$-norm and adopt this attack for AT. Doing so we improve the state-of-the-art in AT-based corruption robustness on CIFAR-10-C. In particular for the contrast and fog corruptions that are known to be problematic (e.g., [25] and [41]), the improvements are a substantial 27% and 13% increase in accuracy.

## 2 Background

We first recall standard adversarial training (AT) with projected gradient descent (PGD). Then we provide background on the BD [10] and the associated mirror descent framework, which generalizes PGD [54].

**Adversarial training.** Let $l(\boldsymbol{x}, y; \theta)$ be a loss of a classifier parameterized by $\theta$ where the input image $\boldsymbol{x} \in [0, 1]^n$ and the label $y$ are sampled from the data distribution $\mathscr{D}$. As formalized by [52], training an adversarially robust model amounts to solving the following min-max optimization problem:

$$\min_{\theta} \mathbb{E}_{(\boldsymbol{x}, y) \sim \mathscr{D}} \left[ \max_{\boldsymbol{x}' \in \mathbb{S}(\boldsymbol{x})} l(\boldsymbol{x}', y; \theta) \right] \tag{1}$$

where $\mathbb{S}(\boldsymbol{x})$ is the set of images that are considered similar to $\boldsymbol{x}$. Under the common $L^p$ threat model, $\mathbb{S}(\boldsymbol{x})$ is defined as an $L^p$ ball centered on $\boldsymbol{x}$ of chosen radius $\epsilon$: $\mathbb{S}(\boldsymbol{x}) = \mathbb{B}(\boldsymbol{x}, \epsilon)$. [1] In this case, the inner maximization problem is solved by PGD, which consists of iterating over two steps: a gradient-based update followed by a projection into $\mathbb{B}(\boldsymbol{x}, \epsilon)$.

**Bregman divergence (BD).** For a strongly convex $h : \mathcal{X} \to \mathbb{R}$ (called base function) on a given space $\mathcal{X}$ (called the primal space) with thus strictly monotonous gradient $\nabla h : \mathcal{X} \to \mathcal{Z}$ ($\mathcal{Z}$ is called the dual space), the associated BD [10] $D_h : \mathcal{X} \times \mathcal{X} \to [0, \infty)$ from $\boldsymbol{x}$ to $\boldsymbol{x}'$ is defined as

$$D_h(\boldsymbol{x}' \parallel \boldsymbol{x}) = h(\boldsymbol{x}') - h(\boldsymbol{x}) - \langle \nabla h(\boldsymbol{x}), \boldsymbol{x}' - \boldsymbol{x} \rangle. \tag{2}$$

The BD is similar to a metric or distance (non-negative, zero iff $\boldsymbol{x} = \boldsymbol{x}'$), except that in general it is not symmetric in its arguments and only satisfies a weaker version of the triangle inequality (whose

---

[1] All threat models add another condition to ensure that the adversarial example $\boldsymbol{x}'$ does not exceed its natural range of pixels.

Table 1: Notation and context of our approach. First column: generic concepts associated with the BD and mirror descent. Second and third column: known instantiations. Last column: our learned BDs with a novel approach to robustness as application.

| Generic | Euclidean norm | KL divergence | Ours |
|---|---|---|---|
| Some space $\mathcal{X}$ | Euclidean space | Discrete distributions | Images |
| Base function $h : \mathcal{X} \to \mathbb{R}$ *(strongly convex)* | $h(\boldsymbol{x}) = \frac{1}{2}\|\boldsymbol{x}\|_2^2$ | $h(\boldsymbol{p}) = \sum_i \boldsymbol{p}_i \log(\boldsymbol{p}_i)$ *(Shannon entropy)* | $h = $ learned $\phi$ *(an input convex NN)* |
| Mirror map $\nabla h : \mathcal{X} \to \mathcal{Z}$ *(strictly monotone)* | $\nabla h(\boldsymbol{x}) = \boldsymbol{x}$ | $\nabla h(\boldsymbol{p})_i = \log(\boldsymbol{p}_i)$ | $\Psi \approx \nabla h$ (approximate gradient) |
| Inverse map $(\nabla h)^{-1} : \mathcal{Z} \to \mathcal{X}$ | $(\nabla h)^{-1}(\boldsymbol{z}) = \boldsymbol{z}$ | $(\nabla h)^{-1}(\boldsymbol{z})_i = e^{\boldsymbol{z}_i}$ | Fenchel conjugate $\overline{\Psi}$ |
| **Bregman Divergence** $D_h(\boldsymbol{x}' \parallel \boldsymbol{x})$ | $\frac{1}{2}\|\boldsymbol{x}' - \boldsymbol{x}\|_2^2$ | $\sum_i \boldsymbol{q}_i \log \frac{\boldsymbol{q}_i}{\boldsymbol{p}_i}$ | $D_\phi$ (learned divergence) |
| **Mirror descent** | PGD | Hedge algorithm | Ours |
| $\boldsymbol{z}^t = \nabla h(\boldsymbol{x}^t)$ $\boldsymbol{z}^{t+1} = \boldsymbol{z}^t - \eta\nabla f(\boldsymbol{x}^t)$ $\boldsymbol{x}^* = (\nabla h)^{-1}(\boldsymbol{z}^{t+1})$ $\boldsymbol{x}^{t+1} = \Pi_{\mathbb{K}}(\boldsymbol{x}^*)$ | $\boldsymbol{x}^* = \boldsymbol{x}^t - \eta\nabla f(\boldsymbol{x}^t)$ $\boldsymbol{x}^{t+1} = \Pi_{\mathbb{B}}(\boldsymbol{x}^*)$ | $\boldsymbol{p}_i^* = \boldsymbol{p}_i^t e^{-\eta\boldsymbol{l}_i}$ $\boldsymbol{p}^{t+1} = \Pi_\Delta(\boldsymbol{p}^*)$ | $\boldsymbol{z}^t = \Psi(\boldsymbol{x}^t)$ $\boldsymbol{z}^{t+1} = \boldsymbol{z}^t + \eta\nabla l(\boldsymbol{x}^t)$ $\boldsymbol{x}^* = \overline{\Psi}(\boldsymbol{z}^{t+1})$ $\boldsymbol{x}^{t+1} = \Pi_{\mathbb{S}}(\boldsymbol{x}^*)$ |

exact form is not relevant here). $D_h$ is convex in its first argument but not necessarily in the second [21]. The projection of an $\boldsymbol{x} \in \mathcal{X}$ on a closed and convex set $\mathbb{K} \subseteq \mathcal{X}$ w.r.t. to $D_h$ exists and is unique:

$$\Pi_{\mathbb{K}}(\boldsymbol{x}) = \arg\min_{\boldsymbol{x}' \in \mathbb{K}} D_h(\boldsymbol{x}' \parallel \boldsymbol{x}). \tag{3}$$

The generic concepts are shown in the first column in Tab. 1; the other columns are examples. The squared Euclidean distance is a BD for $h$ chosen as the squared $L^2$-norm. More interestingly, if $h$ is the negative Shannon entropy, the associated BD is the Kullback-Leibler (KL) divergence. Various other divergences have been defined [6, 67, 29].

The *Bregman ball* centered on $\boldsymbol{x}$ with radius $\epsilon$ is then given by

$$\mathbb{B}_h(\boldsymbol{x}, \epsilon) = \{\boldsymbol{x}' \in \mathcal{X} \mid D_h(\boldsymbol{x}' \parallel \boldsymbol{x}) \le \epsilon\}. \tag{4}$$

The ball $\mathbb{B}_h$ is bounded, compact if $\mathcal{X}$ is closed, and convex [21].

**Mirror descent.** Mirror descent [54] is a framework for optimizing functions $f : \mathcal{X} \to \mathbb{R}$ possibly constrained to a feasible convex set $\mathbb{K}$: $\min_{\boldsymbol{x} \in \mathbb{K}} f(\boldsymbol{x})$, given a suitable base function $h$ that defines a BD. Mirror descent requires the gradient $\nabla h$ (called the *mirror map*) and the existence of $(\nabla h)^{-1}$ (called the *the inverse map*). The algorithm is iterative as shown in the first column in Tab. 1. After initializing $\boldsymbol{x}^0$ at any point in $\mathbb{K}$, each iteration $t$ consists of four steps: *(i)* mapping the current point $\boldsymbol{x}^t$ to a point in the dual space $\boldsymbol{z}^t = \nabla h(\boldsymbol{x}^t)$ through the mirror map, *(ii)* taking a gradient step of size $\eta$: $\boldsymbol{z}^{t+1} = \boldsymbol{z}^t - \eta\nabla f(\boldsymbol{x}^t)$, *(iii)* mapping $\boldsymbol{z}^{t+1}$ back to the primal space using the inverse map: $\boldsymbol{x}^* = (\nabla h)^{-1}(\boldsymbol{z}^{t+1})$, *(iv)* projecting $\boldsymbol{x}^*$ into the feasible set $\mathbb{K}$ w.r.t. $D_h$: $\boldsymbol{x}^{t+1} = \Pi_{\mathbb{K}}(\boldsymbol{x}^*)$ with (3).

As shown in Tab. 1, for the Euclidean divergence, mirror descent is exactly PGD. For the KL divergence it becomes the so-called hedge algorithm [26]. In this paper, as sketched in the fourth column, we will learn base functions $h$ that we call $\phi$ and associated divergences $D_\phi$ for common image corruptions and use them for AT.

## 3 Learning a BD

As first main contribution we exploit the theory of BD to derive new similarity measures for images. Namely, we learn a base function $h = \phi$ that satisfies the properties to make $D_\phi$ a divergence. Mathematically, this $\phi$ will play the same role as the Shannon entropy for KL divergence. Formally, the challenge is to learn a $\phi$ with the following properties:

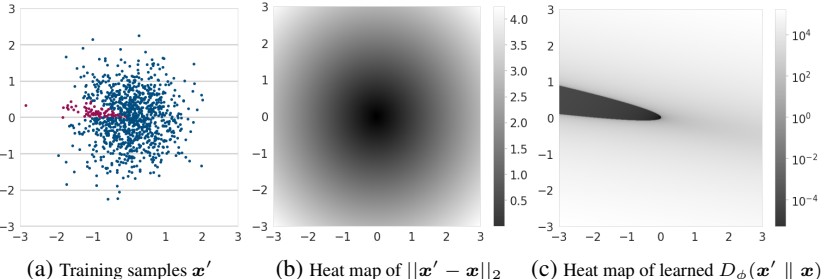

| (a) Training samples $\boldsymbol{x}'$ | (b) Heat map of $\|\boldsymbol{x}' - \boldsymbol{x}\|_2$ | (c) Heat map of learned $D_\phi(\boldsymbol{x}' \parallel \boldsymbol{x})$ |

Figure 1: Learning a BD in two dimensions. (a) The original point is $\boldsymbol{x} = (0, 0)$, the noisy perturbations are in blue, the corrupted points $\tau(\boldsymbol{x})$ (in red) have angles between $\frac{7}{8}\pi$ and $\pi$. (b) Heat map of the $L^2$-distance to the origin, which is unable to distinguish corrupted from noisy points. (c) Heat map of our learned BD trained on the samples in (a), which considers corrupted points very close compared to noisy points.

1. $\phi$ is strongly convex and differentiable, and thus $D_\phi$ a divergence;
2. $\nabla\phi(\boldsymbol{x})$ and $(\nabla\phi)^{-1}(\boldsymbol{x})$ are (approximately) computable to execute mirror descent.

## 3.1 Strongly convex architecture

We propose to model $\phi$ as a deep neural network with a particular architecture: the *input convex neural network (ICNN)* [1, 42] for which we propose a self-supervised learning algorithm. The architecture is an $L$-layered deep neural network with activations $\boldsymbol{z}^l$ defined as:

$$\begin{cases} \boldsymbol{u}^1 = q^0\left[\boldsymbol{W}^0\boldsymbol{x}\right] \\ \boldsymbol{z}^1 = g^0\left[\boldsymbol{U}^1\boldsymbol{u}^1 + \boldsymbol{V}^0\boldsymbol{x} + \boldsymbol{b}^0\right] \\ \boldsymbol{u}^l = q^{l-1}\left[\boldsymbol{W}^{l-1}\boldsymbol{x}\right] \\ \boldsymbol{z}^l = g^{l-1}\left[\boldsymbol{U}^l\boldsymbol{u}^l + \boldsymbol{V}^{l-1}\boldsymbol{z}^{l-1} + \boldsymbol{b}^{l-1}\right] \text{ for } 2 \le l \le L. \end{cases} \tag{5}$$

And finally the output is defined as $\phi(\boldsymbol{x}) = \boldsymbol{z}^L + \frac{\alpha}{2}\|\boldsymbol{x}\|_2^2$ with $\alpha > 0$. The weights $\boldsymbol{W}^l$, $\boldsymbol{U}^l$, and $\boldsymbol{V}^l$ with the biases $\boldsymbol{b}^l$ are learnable parameters while $q^l$ and $g^l$ are non-linear activation functions.

The function $\phi$ is convex provided that all $\boldsymbol{V}^1, .., \boldsymbol{V}^{L-1}$ and $\boldsymbol{U}^1, .., \boldsymbol{U}^{L-1}$ are non-negative and all the activation functions $q^l$ and $g^l$ are convex and non-decreasing [1, Proposition 1]. Furthermore, adding the term $\frac{\alpha}{2}\|\boldsymbol{x}\|_2^2$ to the final layer ensures that $\phi$ is $\alpha$-strongly convex.

We can choose the activations $q^l$ to be the Hadamard square and the weights $\boldsymbol{U}^1, .., \boldsymbol{U}^{L-1}$ to be the identity matrix. As we intend to compute the derivative of this network with respect to the input (to obtain $\Psi$), the derivative of the Hadamard square will be linear feedthroughs. This activation function has proven to be the effective in practical settings. Further, we set all the activation functions $g^l$ to be the continuously differentiable exponential linear unit (CELU) [5] and the linear layers as convolutions. Once we have $\phi$, we numerically approximate the evaluation of the mirror map $\Psi(\boldsymbol{x}) \approx \nabla\phi(\boldsymbol{x})$ using automatic differentiation [58] to obtain the associated divergence as

$$D_\phi(\boldsymbol{x}' \parallel \boldsymbol{x}) = \phi(\boldsymbol{x}') - \phi(\boldsymbol{x}) - \langle\Psi(\boldsymbol{x}), \boldsymbol{x}' - \boldsymbol{x}\rangle. \tag{6}$$

## 3.2 Training divergences for corruptions

A real-world corruption of an image $\tau(\boldsymbol{x})$ (like blurred or with changed contrast) typically lies at a large $L^2$ distance $\epsilon$ (say 10) of the clean image $\boldsymbol{x}$ and thus an $L^2$-based attack with this $\epsilon$ would not find it but instead an extremely noisy one $\tilde{\boldsymbol{x}}$ at similar distance which would likely not be recognizable by a human. As an additional problem, the $L^p$-based AT also does not converge for large $\epsilon$ and typically very small $\epsilon$ around $0.1$ are used [33, 25, 74, 39, 41].

Our second main contribution is to train $\phi$ such that the induced $D_\phi$ considers a corrupted image $\tau(\boldsymbol{x})$ close to the clean $\boldsymbol{x}$ while considering noisy images $\{\tilde{\boldsymbol{x}}^i\}_{i=1}^m$ far away even when the Euclidean

distance suggests the opposite. This means each of the divergences $D_\phi(\tilde{\boldsymbol{x}}^i \parallel \boldsymbol{x})$, $i = 1, ..., m$, should be larger than $D_\phi(\tau(\boldsymbol{x}) \parallel \boldsymbol{x})$ or equivalently $-D_\phi(\tau(\boldsymbol{x}) \parallel \boldsymbol{x}) > -D_\phi(\tilde{\boldsymbol{x}}^i \parallel \boldsymbol{x})$. We propose the following *Bregman loss* $l_B(\boldsymbol{x}; \phi, \Psi)$ to jointly enforce these $m$ inequalities:

$$l_B(\boldsymbol{x}; \phi, \Psi) = -\log \frac{e^{-D_\phi(\tau(\boldsymbol{x})\parallel\boldsymbol{x})}}{e^{-D_\phi(\tau(\boldsymbol{x})\parallel\boldsymbol{x})} + \sum_i e^{-D_\phi(\tilde{\boldsymbol{x}}^i\parallel\boldsymbol{x})}}.$$

The loss $l_B(\boldsymbol{x}; \phi, \Psi)$ can be interpreted as a cross entropy where the logits vector is the negative of the BDs $\left[-D_\phi(\tau(\boldsymbol{x}) \parallel \boldsymbol{x}), -D_\phi(\tilde{\boldsymbol{x}}^1 \parallel \boldsymbol{x}), ..., -D_\phi(\tilde{\boldsymbol{x}}^m \parallel \boldsymbol{x})\right]$ and the ground truth class always corresponds the first entry. Then, we learn $\phi$ by minimizing:

$$\min_{\phi, \Psi} \mathbb{E}_{\boldsymbol{x} \sim \mathcal{D}} \left[l_B(\boldsymbol{x}; \phi, \Psi)\right]. \tag{7}$$

After successful training the Bregman ball $\mathbb{B}_\phi(\boldsymbol{x}, D_\phi(\tau(\boldsymbol{x}) \parallel \boldsymbol{x}))$ contains the transformed image $\tau(\boldsymbol{x})$ by definition but does not contain any of the noisy images $\{\tilde{\boldsymbol{x}}^i\}_{i=1}^m$. We execute this approach on an example in two dimensions as illustrated in Fig. 1.

**Sampling noisy images.** To train for (7) we need a way to sample random images $\{\tilde{\boldsymbol{x}}^i\}_{i=1}^m$ at a distance proportional to that of the corrupted image $||\tau(\boldsymbol{x}) - \boldsymbol{x}||_2$. This distance is controlled by the proportion coefficient $d \in (0, 1]$. In other words, we sample $\{\tilde{\boldsymbol{x}}^i\}_{i=1}^m$ from some distribution $\tilde{x}$ such that:

$$\frac{1}{m} \sum_i ||\tilde{\boldsymbol{x}}^i - \boldsymbol{x}||_2 = d \, ||\tau(\boldsymbol{x}) - \boldsymbol{x}||_2.$$

We chose this distribution to be the isotropic Gaussian:

$$\tilde{x} = \boldsymbol{x} + (1/\sqrt{n-1})d \, ||\tau(\boldsymbol{x}) - \boldsymbol{x}||_2 \boldsymbol{\delta}, \boldsymbol{\delta} \sim \mathcal{N}(0, \boldsymbol{I}_n). \tag{8}$$

This way the expectation $\mathbb{E}\left[||\tilde{x} - \boldsymbol{x}||_2\right]$ is asymptotically equivalent to $d||\tau(\boldsymbol{x}) - \boldsymbol{x}||_2$ (proof in Appendix A).

## 4   Mirror descent adversarial training

As the third main contribution, we use our learned BDs $D_\phi$ to achieve corruption robustness through AT. First, as part of the threat model we define the neighborhood of a clean image $\boldsymbol{x}$ as a Bregman ball:

$$\mathbb{S}(\boldsymbol{x}) = \mathbb{B}_\phi(\boldsymbol{x}, \epsilon). \tag{9}$$

Then, we perform the attack by instantiating mirror descent (Tab. 1) to solve the inner maximization problem in (1). As explained in Sec. 2, doing so requires the inverse map $(\nabla\phi)^{-1}$ and a projection w.r.t. $D_\phi$ that we discuss next.

**Inverse map.** Since $\Psi$ is a gradient of a neural network, its inverse $\Psi^{-1}$ is not readily available. To obtain an approximation, we leverage the Fenchel conjugate [24] $\overline{\phi} : \mathcal{Z} \to \mathbb{R}$ of $\phi$, which exists for convex $\phi$, is again convex, and defined as:

$$\overline{\phi}(\boldsymbol{z}) = \max_{\boldsymbol{x}} \langle \boldsymbol{x}, \boldsymbol{z} \rangle - \phi(\boldsymbol{x}). \tag{10}$$

If $\phi$ is of so-called *Legendre type* (i.e., proper closed, essentially smooth and essentially strictly convex [63]), then [24] states that $(\nabla\phi)^{-1} = \nabla\overline{\phi}$. In general, checking that a function is Legendre type is difficult [6], in particular in this case where the function is a neural network. So instead of deriving a closed-form solution using this result, we use it to motivate an approximation: first defining the conjugate $\overline{\phi}$ again as an ICNN with the exact same architecture as $\phi$ in (5); then training by minimizing: [2]

$$\min_{\overline{\phi}, \overline{\Psi}} \mathbb{E}_{\boldsymbol{x} \sim \mathcal{D}} \left[||\overline{\Psi}(\Psi(\boldsymbol{x})) - \boldsymbol{x}||_2\right]. \tag{11}$$

Now $\overline{\Psi}(\boldsymbol{x}) \approx \nabla\overline{\phi}(\boldsymbol{x})$ is again computed using automatic differentiation and approximates $(\nabla\phi)^{-1}(\boldsymbol{x})$ as desired.

---

[2]In this expression, $\overline{\Psi}$ is not an explicit neural network but rather a gradient of the neural network $\overline{\phi}$ computed w.r.t. the input.

**Projection.** The projection w.r.t. a BD into a general convex set is difficult to compute [19]. Numerical solutions only exist for special sets such as hyperplanes or affine spaces that are not applicable to our set of interest $\mathbb{S}(\boldsymbol{x})$. So to approximate the projection of $\boldsymbol{x}^*$ into $\mathbb{S}(\boldsymbol{x})$ (see last row last column of Tab. 1), we perform a binary search over the segment having $\boldsymbol{x}$ and $\boldsymbol{x}^*$ as endpoint until we find a point $\boldsymbol{x}^{t+1} \in \mathbb{S}(\boldsymbol{x})$. This heuristic is not guaranteed to produce optimal results, as there may exist points $\boldsymbol{x}' \in \mathbb{S}(\boldsymbol{x})$, closer to $\boldsymbol{x}^*$ than $\boldsymbol{x}^{t+1}$, that are out of the considered line segment. However, as we will show, it is fast enough to be incorporated in training and it yields good results (see Sec. 6).

## 5 Related work

**Corruption robustness via data augmentation.** Much of the prior literature on corruption robustness aims to improve out-of-distribution generalization by using simulated and augmented images for training. Many such data augmentation techniques are based on creating synthetic training examples through mixing pairs of training images and their labels. This is achieved for example by linear weighted blending of images [79] or by cutting and pasting parts of an image onto another [77]. Researchers also fused images based on masks computed through frequency spectrum analysis [30], based on adaptive masks [48] or based on model-generated features [70]. Other works considered a hybrid version of these mixing methods [57], a stochastic version of them [57], an ensemble of them [76] or a concurrent combination of them [47].

**Adversarial attacks without $L^p$-norms.** Another line of work focuses on adversarial image perturbations not constrained by $L^p$-norms. [35] introduces semantic adversarial attacks that target image transformation parameters instead of image pixels. Similarly, [22] targets spatial transformations. [34] manipulates the hue and saturation components in the hue saturation value (HSV) color space to create adversarial examples. In addition to colorization, [8] also tweaked the texture of objects within images. [65] modified colors within the invisible range. Some works altered the semantic features of images through conditional generative models [38] or conditional image editing [59].

**Robustness via learned similarity metric.** The closest related work adopts the so-called learned perceptual image patch similarity (LPIPS) to study robustness. LPIPS is a weighted sum of the $L^2$ of the feature maps taken from the activation layers of a trained convolutional network:

$$\text{LPIPS}(\boldsymbol{x}, \boldsymbol{x}') = \sum_l w_l ||\omega_l(\boldsymbol{x}) - \omega_l(\boldsymbol{x}')||_2, \tag{12}$$

where $\omega_l$ is the feature map up to the $l$-layer and $w_l$ weighs the contribution of the layer $l$. [71] and [51] propose an attack similar to [11] by adding the LPIPS along with the $L^p$-norm. Differently, [41] and [46] used LPIPS as a function to define the set of similar images (refer to Sec. 2 for notation context): $\mathbb{S}(\boldsymbol{x}) = \{\boldsymbol{x}' \in \mathbb{R}^n | \text{LPIPS}(\boldsymbol{x}, \boldsymbol{x}') \leq \epsilon\}$. Since the projection into this LPIPS-based set does not admit a closed-from expression, solving the inner maximization problem of (1) (i.e., performing the adversarial attack) requires approximation [46] or relaxation [41]. The resulting attacks and their associated AT have been proven effective to train robust models against common image corruptions. We compare against LPIPS in our experiments.

**Learning BDs.** BDs have been widely used in machine learning but are typically hand-engineered and not learned [4, 73, 44]. [66, 17, 62] learn BDs relying on piecewise linear functions and linear lower bound approximation to ensure the convexity of the learned base functions. These methods are limited to low-dimensional inputs, either tabular data or extracted features. [66] uses Gorubi solvers [28] for lower bounds approximation as part of clustering/ranking algorithms. Similarly, [17] use functional BD and apply it to clustering while [62] learn the architecture proposed by [66] through a contrast learning algorithm. Recently, [50] proposed to learn a BD for clustering where its input are features extracted from a CNN. In contrast, we are the first to learn an end-to-end BD on images from raw data where the inputs to the divergence are pixels in a way that yields a convex base functions by construction without bound approximation. This allows us to instantiate the Bregman ball (to define robustness) and to run the mirror descent framework (to train for robustness), which is not possible with prior methods.

## 6 Corruption-specific Bregman divergences

We first show that we can successfully learn a BD that assesses corrupted images (for a given type of corruption) as close and randomly perturbed ones as far from the clean image, even if in Euclidean

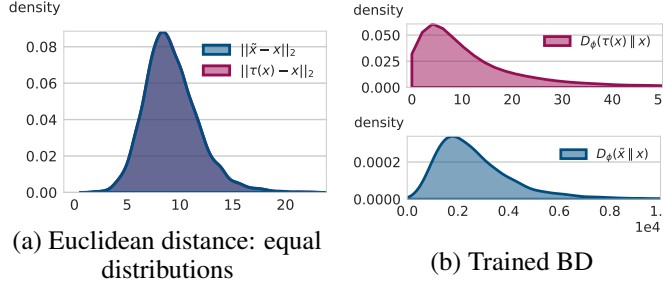

(a) Euclidean distance: equal distributions

(b) Trained BD

Figure 2: (a) Noisy (blue) and contrast-corrupted (red) images chosen to have equal distribution in Euclidean distance to the clean image, and (b) the associated distributions of the learned BDs. Done over 10,000 CIFAR-10 test set images.

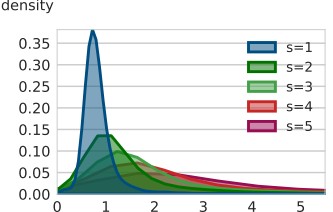

Figure 3: Distribution of trained BDs for contrast-corrupted images $D_\phi(\tau(\boldsymbol{x}) \parallel \boldsymbol{x})$ with multiple severities over 10,000 CIFAR-10 test set images.

distance the converse holds (see Sec. 3). We perform experiments on CIFAR-10 [43] and consider the 14 noise-free corruptions from CIFAR-10-C [33] that can be applied with severities from 1 to 5. One focus are the corruptions of contrast and fog, which have been found the most challenging in AT [25, 41]. We first analyze the learned BDs and then show robustness results when used with AT.

## 6.1 Learning the BD

Learning a BD amounts to learning the base function. For both the base function $\phi$ and its conjugate $\overline{\phi}$ we use the same architecture: an ICNN with 12 convolutional layers followed by 4 fully connected layers. The strong-convexity parameter is chosen as $\alpha = 10^{-3}$. The mirror map and the inverse map are numerically approximated using `autograd.grad` from PyTorch's automatic differentiation engine [58]. As an initialization phase, we first train $\phi$ and $\overline{\phi}$ such that $\Psi$ and $\overline{\Psi}$ approximate the identity function (so initially $\overline{\Psi} = \Psi^{-1}$ holds) on uniformly drawn samples from the usual range of pixels $[0, 1]^n$:

$$\min_{\phi, \Psi} \mathbb{E}_{\boldsymbol{x} \sim \mathcal{U}([0,1]^n)} \left[ ||\Psi(\boldsymbol{x}) - \boldsymbol{x}||_2 \right]. \tag{13}$$

This identity training is performed for 7,000 steps using the Adam optimizer [40] with a batch size of 64, a learning rate of $3 \cdot 10^{-4}$ and no weight decay. For a given corruption $\tau$, we then train $\phi$ with (7) while randomly sampling its severity (1 to 5) for each image at each epoch. The hyperparameter $d$ for sampling noisy images in (8) is uniformly sampled from $[10^{-7}, 0.99]$. The training batch contains 32 clean images, one corrupted image for each clean image, and $m = 63$ samples of noisy images per clean image (2,080 images in total). The training is performed for 10 epochs using the AdamW optimizer [49] with an initial learning rate of $10^{-4}$ and a weight decay of $10^{-9}$. After each update of $\phi$ according to (7), we also update $\overline{\phi}$ according to (11). Finally, we freeze the parameters of $\phi$ and continue training $\overline{\phi}$ for an additional 10 epochs. The training converged for 10 out of the 14 considered corruptions.

It is conceivable to train a BD to be symmetric, however, it is not a good idea since a perfectly symmetric BD is just a quadratic function. However, we found that our learned divergence is qualitatively symmetric in the sense that it performs equally well with flipped arguments (see App. D for details).

**Performance on corruption vs. noise.** We first show in Fig. 2 that the learned divergence $D_\phi$ on images (dimension $n = 3072$) agrees with the 2D example in Fig. 1. To do so we consider, for the test set of 10,000 clean images $\boldsymbol{x}$, contrast-corrupted images $\tau(\boldsymbol{x})$ with severity $s = 5$ (red, one per clean image) and a set of noisy images $\tilde{\boldsymbol{x}}$ (blue, one per clean image). The noisy images are sampled from (8) with $d = 1.0$. With this choice, the distribution of the $L^2$-distances to the clean image is equal for the noisy and the corrupted images (Fig. 2.a). Fig. 2.b shows the distribution of learned divergences to the clean image. Here, all corrupted images are very close (mean 3.8, std 6.0) but the noisy ones far (mean 8385, std 4939), which shows that the learned BD works as intended. Visual results on images from ImageNet are provided in App. D.

Next, we generate multiple corrupted images with different severities from $s = 1$ to $s = 5$ and report their divergences from the clean images in Fig. 3. The divergence considers more severely corrupted

images further from the clean images as expected. All these results are qualitatively the same for all 10 corruptions with learned BDs.

**Comparison against other similarity measures.** We evaluate how well different similarity measures distinguish between noisy and corrupted images. For each clean image and the corresponding corrupted version $\tau(\boldsymbol{x})$ with severity 5, we sample 9 noisy images $\tilde{\boldsymbol{x}}$. We repeat the sampling for different noise coefficients $d$ as shown in Fig. 4. We consider 5 similarity measures $\delta$ to distinguish between noisy and corrupted images: $\delta = L^2$ over the pixels, our trained BD $\delta = D_\phi$, and the three state-of-the-art metric learning methods Dino [13], Unicom [2], and Moco (v2) [15].

First we measure the similarities: $\delta(\tau(\boldsymbol{x}), \boldsymbol{x}), \delta(\tilde{\boldsymbol{x}}^1, \boldsymbol{x}), \ldots, \delta(\tilde{\boldsymbol{x}}^9, \boldsymbol{x})$. An accurate similarity measure yields $\delta(\tau(\boldsymbol{x}), \boldsymbol{x})$ smaller than the rest. We test this accuracy over the test set for multiple values of noise coefficients $d$ in Fig. 4a. Our learned divergence performs best by far, and considers noisy images further compared to corrupted images for all $d$, whereas other state-of-the-art metric learning measures only do so for high noise $d$.

Next, we inspect the ratio $r = \delta(\tilde{\boldsymbol{x}}, \boldsymbol{x})/\delta(\tau(\boldsymbol{x}), \boldsymbol{x})$ and report the aggregated results over the test set in Fig. 4b. For $\delta = L^2$ this ratio is $d$ by construction. We observe that all learned $\delta$ offer better ratios (distinguish corrupted from noisy images better) than $L^2$ and that this distinction improves with $d$ as expected. Consistent with the accuracy results, our trained BD outperforms the other measures by yielding ratios $r > 1$ for all noise levels $d$,

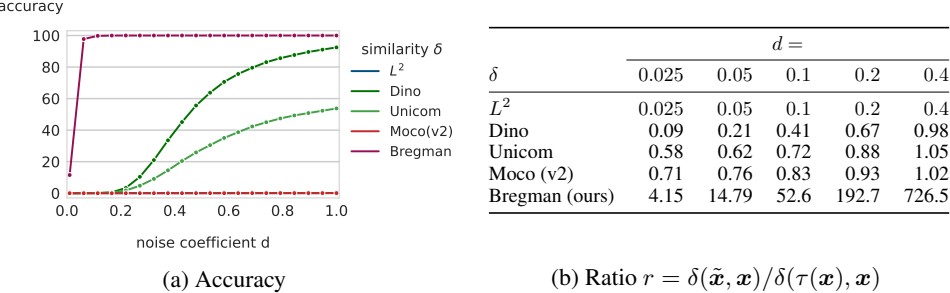

| (a) Accuracy | (b) Ratio $r = \delta(\tilde{\boldsymbol{x}}, \boldsymbol{x})/\delta(\tau(\boldsymbol{x}), \boldsymbol{x})$ |

Figure 4: Comparing the similarity of corrupted images $\delta(\tau(\boldsymbol{x}), \boldsymbol{x})$ against the similarity of noisy images $\delta(\tilde{\boldsymbol{x}}, \boldsymbol{x})$ considering different similarity measures $\delta$, different noise levels $d$, averaged over the test set. We test whether $\delta(\tau(\boldsymbol{x}), \boldsymbol{x}) < \delta(\tilde{\boldsymbol{x}}, \boldsymbol{x})$ and report it as an accuracy in (a). In (b), we further inspect the ratio $\delta(\tilde{\boldsymbol{x}}, \boldsymbol{x})/\delta(\tau(\boldsymbol{x}), \boldsymbol{x})$ that should be $> 1$ for a successful noise-corruption distinction.

## 6.2 AT with mirror descent

As an application of our learned BD, we perform AT by instantiating the associated mirror descent (see Sec. 4) and compare against the relaxed LPIPS AT (RLAT) [41]. We show that the proposed method improves the state of the art in adversarial training-based robustness on several common image corruptions. For the classification model $f$, we use the PreAct ResNet-18 architecture [32], which was also used by [41]. For a fair comparison, we set the number of iterations for our attack to $T = 1$ to match the one-step attack used in RLAT. We also compare against the $L^2$ PGD AT. AT details are reported in App E. The mirror descent is executed following Alg. 2. Samples from the generated adversarial images are provided in App. E.

**AT for contrast corruption.** Both PGD and RLAT fail to improve robustness against contrast as found by [25] and [41] and replicated in Tab. 2. Our mirror descent AT using the learned BD for contrast improve this robustness considerably across all severities (on average from 63.92% to 87.4%). Surprisingly, mirror descent AT for the zoom blur corruption yields further improvement to 90.03% on average. We discuss the reason in the next expanded experiment.

**Comparing AT for different corruptions.** We expand the previous experiment by mirror descent AT for fog, and brightness corruptions and report the average accuracy across severities in Tab. 6 for different corruptions. In all considered cases, our mirror descent AT maintains high accuracy on clean images. We notice that AT with the zoom blur divergence performs best for contrast, brightness, and very well for fog, but, surprisingly, not for zoom blur, for which LPIPS AT is best.

Table 2: Comparison of corruption robustness of models trained under different regimes.

| | | Standard accuracy | Contrast | | | | | Average |
|---|---|---|---|---|---|---|---|---|
| | | | $s=1$ | $s=2$ | $s=3$ | $s=4$ | $s=5$ | |
| | Standard training | 95.12 | 94.52 | 91.32 | 87.50 | 86.40 | 38.68 | 78.13 |
| Adversarial training | PGD | 93.52 | 91.68 | 82.96 | 72.31 | 51.43 | 21.26 | 63.92 |
| | RLAT | 93.27 | 91.47 | 82.32 | 70.65 | 48.35 | 21.58 | 62.87 |
| | Mirror Descent with $D_\phi^{\text{contrast}}$ | 94.04 | 93.99 | 92.77 | 91.20 | 88.33 | 70.72 | 87.40 |
| | Mirror Descent with $D_\phi^{\text{zoom-blur}}$ | **95.16** | **95.00** | **93.94** | **92.75** | **90.41** | **78.05** | **90.03** |

Table 3: Corruption robustness of the standard-trained model against adversarially trained models under $L^2$, RLAT, and our mirror descent (MD) AT for different corruptions.

| | | Standard | Contrast | Fog | Zoom blur | Brightness |
|---|---|---|---|---|---|---|
| | Standard training | 95.12 | 78.13 | 88.72 | 78.86 | 93.46 |
| Adversarial training | PGD | 93.52 | 63.92 | 77.47 | 85.87 | 91.88 |
| | RLAT | 93.27 | 62.87 | 77.00 | **85.88** | 91.72 |
| | MD $D_\phi^{\text{contrast}}$ | 94.04 | 87.40 | 90.34 | 80.81 | 92.51 |
| | MD $D_\phi^{\text{fog}}$ | 94.62 | 83.77 | **90.87** | 81.84 | 93.03 |
| | MD $D_\phi^{\text{zoom-blur}}$ | **95.16** | **90.03** | 90.50 | 79.59 | **93.47** |
| | MD $D_\phi^{\text{brightness}}$ | 94.71 | 88.61 | 90.16 | 79.16 | 93.31 |

We notice a high degree of cross-corruption robustness generalization for the shown models. E.g., a model trained for brightness also performs well on contrast. The reason is that the underlying BDs exhibit the same kind of generalization, i.e., the BD trained for brightness also considers contrast corrupted images as close to the original. We provide a detailed analysis in Appendix D. For the 6 corruptions with learned BDs not shown, our AT did not improve over prior work which we attribute to the small scale of our ICNN; see limitations below.

# 7 Corruption-oblivious Bregman divergence

Next we train a BD on a distinct dataset of varying corruptions, thereby rendering it oblivious to CIFAR-10-C. Specifically, we use the Berkeley-Adobe Perceptual Patch Similarity (BAPPS) data set [81], which is a collection of image triplets (reference, distortion 1, distortion 2) and a human judgment stating which of distortions is similar to the reference (so no classification labels). The data set features a diverse range of images and distortions, spanning 6 categories, including traditional and CNN-based distortions. The former modify images through a combination of low-level edits such as saturation adjustments and spatial warping, while CNN-based distortions alter the parameters of a generative model to produce distorted images. We show results on BD learning and for robustness with AT on this data set.

## 7.1 Learning the BD

Somewhat different from the BD learning before, we train here the BD to mimic the human judgment in BAPPS, evaluated on the 2AFC test, which provides 6 categories of test data. To do so we consider images human-judged to be more similar as close, and the other one as far from the original. Since we are not using additional pre-training data, we compare against the VGG version of LPIPS that does not use ImageNet pre-training. The training pipeline (loss, optimizers, batch size, etc.) is similar to that in [81]. We report the accuracy results on the 6 test categories of 2AFC in Tab. 4. Our method outperformed LPIPS in all categories except for Frame Interpolation.

## 7.2 AT with mirror descent

We perform again AT by training a BD on the entire BAPPS, different from Sec. 7.1, as described in Sec. 3.2 except that the corrupted image is not generated by a corruption $\tau$ but rather it is one of the distortions from BAPPS. This BD is thus oblivious to any particular corruption and those in

Table 4: Accuracy of the trained Bregman divergence compared to LPIPS evaluated on different categories of the 2AFC task from the BAPPS dataset.

| | Traditional | CNN-based | Super Resolution | Video deblur | Colorization | Frame Interpolation |
|---|---|---|---|---|---|---|
| LPIPS | 51.41 | 72.10 | 60.46 | 54.25 | 55.18 | **55.55** |
| Bregman (ours) | **63.65** | **79.57** | **61.04** | **56.95** | **61.63** | 53.73 |

Table 5: Corruption robustness of the learned corruption-oblivious Bregman divergence compared to PGD and RLAT.

| | Clean | Contrast | Fog | Zoom blur |
|---|---|---|---|---|
| PGD | **93.65** | 63.19 | 77.18 | 86.08 |
| RLAT | 93.28 | 62.87 | 77.01 | 85.89 |
| Bregman (ours) | 93.61 | **77.70** | **88.00** | **87.12** |

CIFAR-10-C. We then re-execute adversarial training on CIFAR-10 with this divergence. The results on CIFAR-10-C in Tab. 5 show again that our method outperforms RLAT and PGD especially for the fog and the contrast corruptions where PGD and RLAT are known to fail [25] and [41]. Zoom blur improves over all prior results in Tab. 6. Again our AT does not improve the other categories.

## 8 Discussion

**Limitations.** When used with AT, our method introduces an overhead in first training for a valid divergence and then in performing the mirror descent with the heuristic projection (see App. C for a detailed cost analysis). Our method does provide adversarial examples within the trained Bregman ball using the suggested line search projection, however a better heuristic for projection is one important avenue for improvements.

The training of BDs did not succeed for some CIFAR-10-C corruptions nor for all its corruptions simultaneously, but worked on BAPPS. Further, AT with our mirror descent-inspired AT is unable to improve robustness on prior work for several corruptions. We attribute these issues to the small scale of the used convex architecture $\phi$. Scaling up and training on larger data sets with larger image sizes should be easily straightforward with more GPUs, instead of the one V100 GPU we had access to. However, despite the relatively small scale, the results in Sec. 7 demonstrate that our method can successfully learn a corruption-oblivious BD that exhibits robust generalization across various types of corruptions, when given a suitable training set.

**Broader impact.** One of the contributions of this paper is to increase the corruption robustness of machine vision models specifically by using the AT machinery. Corruption robustness enhances the reliability and safety of models deployed in various applications such as autonomous driving.

## 9 Conclusion

We see our main contribution in showing how to learn a BD from raw high-dimensional data with an approach that should generalize to settings other than the image corruptions considered here. The benefit is in importing the associated theoretical underpinning of the BD such as the compactness of Bregman balls and the well-established mirror descent. The latter motivated us to consider AT for corruption robustness as prototypical application. We considered the two very different data sets CIFAR-10-C and BAPPS to obtain both corruption-specific and corruption-oblivious BDs, and demonstrated that they are consistent in various ways: the former approximately symmetric and monotonous in the corruption severity, the latter outperforming LPIPS in mimicking human judgment. The associated ATs were particularly successful on contrast and fog that troubled prior work.

Our contribution is only a first step and opens various avenues for further improvements including the use of more complex architectures for learning the base functions and thus the BDs, better heuristics for the mirror descent projections, and applications and data sets beyond images.

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

# A   Asymptotic equivalence of noisy image sampling expectation

This section presents the proof mentioned in Sec. 3.2: the expectation $\mathbb{E}\left[||\tilde{\mathrm{x}} - \boldsymbol{x}||_2\right]$ is asymptotically equivalent to $d\,||\tau(\boldsymbol{x}) - \boldsymbol{x}||_2$. We first show that:

$$\mathbb{E}[||\tilde{\mathrm{x}} - \boldsymbol{x}||_2] = \frac{\sqrt{2}\Gamma(\frac{n+1}{2})}{\sqrt{n-1}\Gamma(\frac{n}{2})} d\,||\tau(\boldsymbol{x}) - \boldsymbol{x}||_2$$

Then we simplify this expression.

**Deriving the expectation.** Let $\boldsymbol{x} \in \mathbb{R}^n$ a fixed image and $\tilde{\mathrm{x}}$ a random variable defined as follows with $\mu > 0$:

$$\tilde{\mathrm{x}} = \boldsymbol{x} + \mu\boldsymbol{\delta}\,,\ \boldsymbol{\delta} \sim \mathcal{N}(0, \boldsymbol{I}_n), \tag{14}$$

The random variable $\tilde{\mathrm{x}}$ is a gaussian because it is a linear combination of gaussians ($\boldsymbol{x}$ is fixed).

$$
\begin{aligned}
||\tilde{\mathrm{x}} - \boldsymbol{x}||_2 &= \sqrt{\sum_i (\tilde{\mathrm{x}}_i - \boldsymbol{x}_i)^2} \\
&= \mu\sqrt{\sum_i \boldsymbol{\delta}_i^2} \\
&= \mu\sqrt{\mathrm{z}}
\end{aligned}
\tag{15}
$$

$\mathrm{z}$ is a Chi-square distribution of degree $n$ with density:

$$p_{\mathrm{z}}(z) = \frac{z^{n/2-1}e^{-z/2}}{2^{n/2}\Gamma(\frac{n}{2})} \tag{16}$$

We defined the variable $\mathrm{u} = f(\mathrm{z}) = \sqrt{\mathrm{z}}$, $\mathrm{u} \geq 0$. The density of $\mathrm{u}$ can be computed by the change or variable formula:

$$
\begin{aligned}
p_{\mathrm{u}}(u) &= p_{\mathrm{z}}(f^{-1}(u))\left|\frac{dz}{du}\right| \\
&= \frac{u^{n-1}e^{-u^2/2}}{2^{n/2-1}\Gamma(\frac{n}{2})}
\end{aligned}
\tag{17}
$$

Next, we compute the expectation of $\mathrm{u}$:

$$
\begin{aligned}
\mathbb{E}(\mathrm{u}) &= \int_0^\infty u p_{\mathrm{u}}(u)du \\
&= \frac{1}{2^{n/2-1}\Gamma(\frac{n}{2})}\int_0^\infty u^n e^{-u^2/2}du \\
&= \frac{\sqrt{2}}{\Gamma(\frac{n}{2})}\int_0^\infty t^{(n-1)/2}e^{-t}dt \qquad \text{(by substituting } u = \sqrt{2t}) \\
&= \frac{\sqrt{2}}{\Gamma(\frac{n}{2})}\Gamma(\frac{n+1}{2}) \qquad\qquad\qquad \text{(by definition of } \Gamma)
\end{aligned}
\tag{18}
$$

So we have:

$$\mathbb{E}[||\tilde{\mathrm{x}} - \boldsymbol{x}||_2] = \mathbb{E}(\mu\mathrm{u}) = \mu\frac{\sqrt{2}\Gamma(\frac{n+1}{2})}{\Gamma(\frac{n}{2})} \tag{19}$$

In Equ. 14, we set:

$$\mu = \frac{d \, ||\tau(\boldsymbol{x}) - \boldsymbol{x}||_2}{\sqrt{n-1}} \tag{20}$$

to conclude that for the Gaussian defined in (8):

$$\mathbb{E}[||\tilde{\mathbf{x}} - \boldsymbol{x}||_2] = \frac{\sqrt{2}\Gamma(\frac{n+1}{2})}{\sqrt{n-1}\Gamma(\frac{n}{2})} d \, ||\tau(\boldsymbol{x}) - \boldsymbol{x}||_2 \tag{21}$$

**Asymptotic equivalence.** To prove that $\mathbb{E}[||\tilde{\mathbf{x}} - \boldsymbol{x}||_2]$ is asymptotically equivalent to $d \, ||\tau(\boldsymbol{x}) - \boldsymbol{x}||_2$, it suffices to prove that $\Gamma(\frac{n}{2})/\Gamma(\frac{n+1}{2})$ is is asymptotically equivalent to $\sqrt{2}/\sqrt{n-1}$. The later can be obtained using Laplace/Hayman's method following the steps explained by [3]:

$$\frac{\Gamma(\frac{n}{2})}{\Gamma(\frac{n+1}{2})} = \frac{2}{\sqrt{\pi}} \int_0^{\pi/2} (\cos\theta)^{n-1} \, d\theta \sim \frac{2}{\sqrt{\pi}} \int_0^{+\infty} \exp\left[-(n-1)\frac{\theta^2}{2}\right] d\theta = \sqrt{\frac{2}{n-1}}. \tag{22}$$

## B  Pseudo-code of the mirror descent-based AT

In this section, we provide the pseudo-code of two majors phases of our method. First, Alg. 1 is for the training of BD that was discussed in Sec. 3.2 and its inverse map presented in Sec. 4. Our instantiation of the mirror descent procedure used for adversarial training (see Sec. 4) is detailed in Sec.2. In practice, all these training procedures are performed on batches of images but here we present them for one image. We also omit the validation loops and early stopping conditions to improve readability.

---

**Algorithm 1** Self-supervised BD training

---

**Input:** unlabeled training set $\mathscr{D}$, a meaningful image corruption $\tau$ (such as blur).
**Output:** a trained base function $\phi$ and its approximated Fenchel conjugate $\overline{\phi}$.
Pre-train $\phi$ such as its mirror map $\Psi$ fits the identity function following Equ. 13.
Copy the parameters of $\phi$ to $\overline{\phi}$.
**for** $e = 1$ **to** $E$ **do**
  **for** each image $\boldsymbol{x}$ in the training set $\mathscr{D}$ **do**
    Compute the corrupted image $\tau(\boldsymbol{x})$ and the distance $||\tau(\boldsymbol{x}) - \boldsymbol{x}||_2$.
    Sample $\tilde{\boldsymbol{x}}^1, \tilde{\boldsymbol{x}}^2, ..., \tilde{\boldsymbol{x}}^m$.
    Perform a forward pass on $\phi$ to obtain $\phi(\boldsymbol{x}), \phi(\tau(\boldsymbol{x})), \phi(\tilde{\boldsymbol{x}}^1), \phi(\tilde{\boldsymbol{x}}^2), ..., \phi(\tilde{\boldsymbol{x}}^m)$
    Perform a backward pass on $\phi$ to obtain $\boldsymbol{z} = \Psi(\boldsymbol{x})$.
    Compute the Bregman loss $l = -\log \frac{e^{-D_\phi(\tau(\boldsymbol{x})\|\boldsymbol{x})}}{e^{-D_\phi(\tau(\boldsymbol{x})\|\boldsymbol{x})} + \sum_i e^{-D_\phi(\tilde{\boldsymbol{x}}^i\|\boldsymbol{x})}}$.
    Perform an AdamW optimization step [49] on the parameters of $\phi$ and $\Psi$ to minimize $l$.
    Freeze the parameters of $\phi$ and $\Psi$.
    Perform a backward pass on $\overline{\phi}$ to obtain $\boldsymbol{x}' = \overline{\Psi}(\boldsymbol{z})$.
    Compute the MSE loss $l' = ||\boldsymbol{x}' - \boldsymbol{x}||_2^2$.
    Perform an AdamW on the parameters of $\overline{\phi}$ and $\overline{\Psi}$ to minimize $l'$.
    Unfreeze the parameters of $\phi$ and $\Psi$.
  **end for**
**end for**
**return** $\phi, \overline{\phi}$

---

## C  Cost analysis of the Mirror descent adversarial training

Compared to the standard AT, the (multiplicative) overhead is bounded by $O(K)$ where $K$ is the depth of the convex NNs used. In our case $K = 14$. In practice, our method requires about twice the runtime as the standard AT when implemented in PyTorch and run on a single V100 GPU.

**Algorithm 2** Our instantiation of the mirror descent

---

**Input:** clean image $\boldsymbol{x}$, its ground truth label $y$, trained base function $\phi$, mirror map $\Psi$ and inverse map $\overline{\Psi}$
**Output:** potentially misclassified image inside in the Bregman ball $\boldsymbol{x}' \in \mathbb{B}_\phi(\boldsymbol{x}, \epsilon)$.
Initialize $\boldsymbol{x}' = \boldsymbol{x}$.
**for** $t = 1$ **to** $T$ **do**
   $\eta = \epsilon 10^{\frac{-4t}{T}}$
   $\boldsymbol{z}' = \Psi(\boldsymbol{x}')$
   $\boldsymbol{z}' = \boldsymbol{z}' + \eta \nabla_{\boldsymbol{x}} l(\boldsymbol{x}', y; \theta)$
   $\boldsymbol{x}' = \overline{\Psi}(\boldsymbol{z}')$
   $a = 0$
   $b = 1$
   **while** $D_\phi(\boldsymbol{x}' \parallel \boldsymbol{x}) > \epsilon$ **do**
      $m = (a + b)/2$
      $\boldsymbol{x}' = \boldsymbol{x} + m(\boldsymbol{x}' - \boldsymbol{x})$
      **if** $D_\phi(\boldsymbol{x}' \parallel \boldsymbol{x}) > \epsilon$ **then**
         $a = m$
      **else**
         $b = m$
      **end if**
   **end while**
   $\boldsymbol{x}' = \texttt{clip}(\boldsymbol{x}', 0, 1)$
**end for**
**return** $\boldsymbol{x}'$

---

This overhead is due to replacing the computation of the $L^2$ norm by the more costly BD and the application of the mirror map $\Psi$ and its inverse. Computing one BD amounts to two forward passes on the NN $\phi$ and one backward pass on $\phi$ while computing a mirror (or inverse) map is one backward pass on $\phi$ (or $\overline{\phi}$), respectively. The cost of each forward or backward pass is linear in $K = 14$, the depth of the neural network $\phi$ (or $\overline{\phi}$).

## D Further results about the learned divergence

**Trained based function.** In Sec. 6, we have shown that our trained BD is semantically meaningful as it considers noisy image far off clean images and the corrupted images closer even when the $L^2$ says otherwise. Here, we inspect the learned base functions $\phi$, i.e., our trained "entropy" for images. The distribution of its outputs on the test set is shown in Fig. 5. We notice that the trained based function have a modality with an amplitude different than that of the $L^2$-norm, the base function in the $L^2$-based threat models (see Tab. 1).

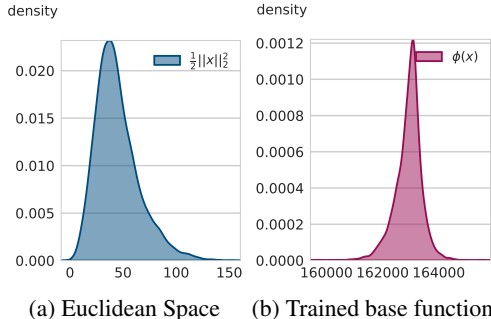

(a) Euclidean Space    (b) Trained base function

Figure 5: Distribution of BD's base functions over 10,000 CIFAR-10 test set images. Our trained base function $\phi$ (b) is compared against (a) its counterpart in the PGD setting, the half of norm $L^2$ squared (see Tab. 1).

**Evaluation on higher dimensions.** The performance of Bregman divergence on higher dimensions matches that presented for lower resolution (32x32) as shown by Figure 6. The Bregman divergence successfully distinguishes corrupted from noisy 256x256 ImageNet images despite being trained on 32x32 CIFAR-10 images.

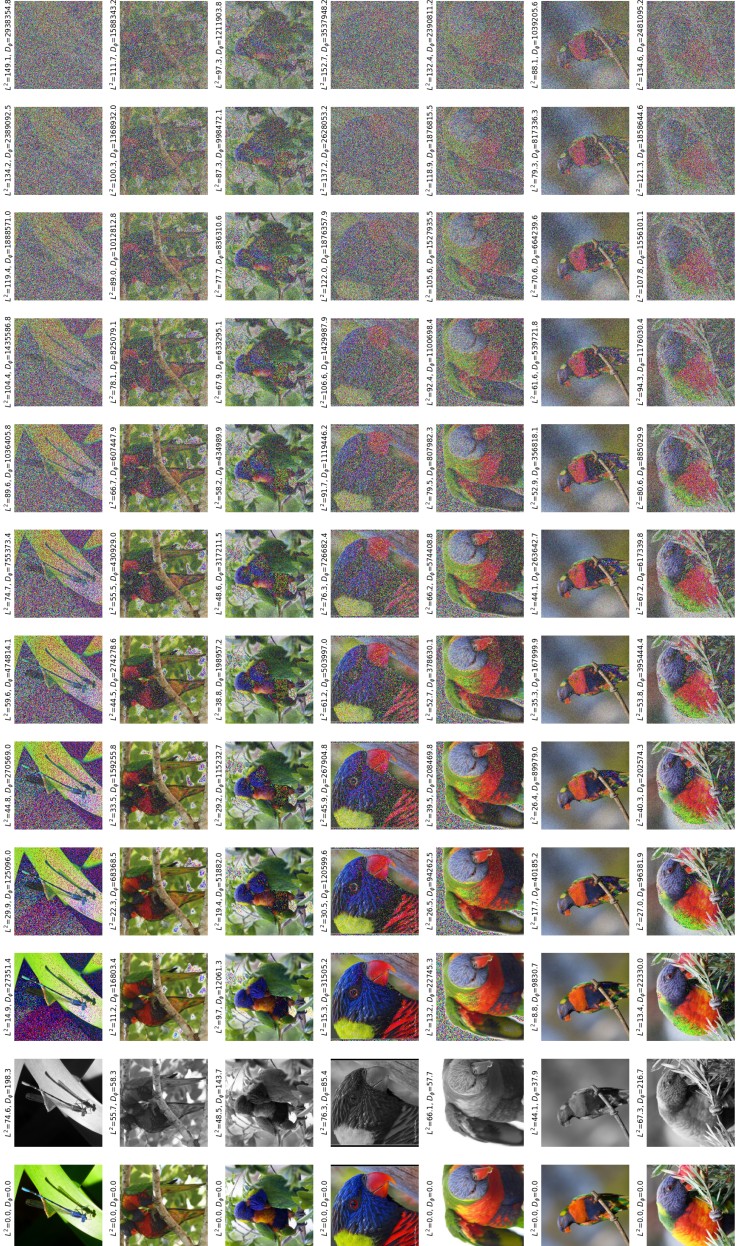

Figure 6: Evaluation of the Bregman divergence of 256x256 images from ImageNet. The clean images are plotted in the first column, corrupted versions in the second column, and noisy versions (with different noise thresholds) thereafter

**Symmetricity.** It is conceivable to train the divergence to be symmetric $D_\phi(\boldsymbol{x}' \parallel \boldsymbol{x}) \approx D_\phi(\boldsymbol{x} \parallel \boldsymbol{x}')$ by minimizing the following loss in conjunction with the Bregman loss in (7): $\mathbb{E}\left[\|D_\phi(\boldsymbol{x}' \parallel \boldsymbol{x}) - D_\phi(\boldsymbol{x} \parallel \boldsymbol{x}')\|_2\right]$. However, this is not a good idea because enforcing the symmetry limits the expressive power of the learned divergence, since a perfectly symmetric $D_\phi$ is just a quadratic function. And indeed, our trained BD is not symmetric as shown in Fig. 7. However, we

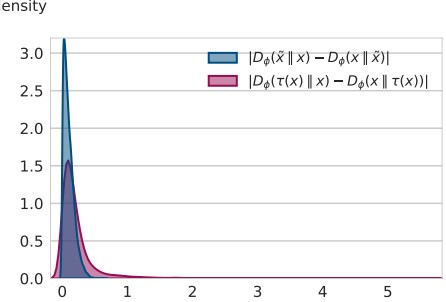

Figure 7: Symmetricity test of the trained BD over the 10,000 images of the test set.

Table 6: Corruption robustness of the standard-trained model against adversarially trained models under $L^2$, RLAT, and our mirror descent (MD) AT for different corruptions.

| | | Standard | Contrast | Fog | Zoom Blur | Brightness |
|---|---|---|---|---|---|---|
| | Standard training | 95.12 | 78.13 | 88.72 | 78.86 | 93.46 |
| Adversarial training | PGD | 93.52 | 63.92 | 77.47 | 85.87 | 91.88 |
| | RLAT | 93.27 | 62.87 | 77.00 | **85.88** | 91.72 |
| | MD $D_\phi^{\text{contrast}}$ | 94.04 | 87.40 | 90.34 | 80.81 | 92.51 |
| | MD $D_\phi^{\text{fog}}$ | 94.62 | 83.77 | **90.87** | 81.84 | 93.03 |
| | MD $D_\phi^{\text{zoom-blur}}$ | **95.16** | **90.03** | 90.50 | 79.59 | **93.47** |
| | MD $D_\phi^{\text{brightness}}$ | 94.71 | 88.61 | 90.16 | 79.16 | 93.31 |

Table 7: Evaluating a BD learned for a corruption $\tau$ ($D_\phi^\tau$ in rows) on different corruptions $\tau'$ (in columns) by computing the ratio $D_\phi^\tau(\tau'(\boldsymbol{x}) \parallel \boldsymbol{x})/D_\phi^\tau(\tau(\boldsymbol{x}) \parallel \boldsymbol{x})$ averaged over the test set.

| | $\tau' =$ | | | |
|---|---|---|---|---|
| $D_\phi^\tau$ | Contrast | Fog | Zoom blur | Brightness |
| $D_\phi^{\text{contrast}}$ | 1.0 | 2.32 | 4.23 | 10.97 |
| $D_\phi^{\text{fog}}$ | 1.64 | 1.0 | 4.84 | 10.46 |
| $D_\phi^{\text{zoom-blur}}$ | 1.17 | 0.89 | 1.0 | 5.34 |
| $D_\phi^{\text{brightness}}$ | 1.11 | 0.68 | 1.03 | 1.0 |

notice that $D_\phi(\boldsymbol{x}' \parallel \boldsymbol{x})$ and $D_\phi(\boldsymbol{x} \parallel \boldsymbol{x}')$ have similar magnitudes, which one can see as a consistency property.

**Cross-corruption generalization.** We consider trained divergences $D_\phi^\tau$ for multiple corruptions $\tau$. As explained in Sec. 6, $D_\phi^\tau$ considers an image and its corrupted version $\tau(\boldsymbol{x})$ as close, i.e., $D_\phi^\tau(\tau(\boldsymbol{x}) \parallel \boldsymbol{x})$ is small. Now, we pick another corruption $\tau'$ and investigate whether $D_\phi^\tau$ also considers $\tau'(\boldsymbol{x})$ close to $\boldsymbol{x}$, by computing the ratio $D_\phi^\tau(\tau'(\boldsymbol{x}) \parallel \boldsymbol{x})/D_\phi^\tau(\tau(\boldsymbol{x}) \parallel \boldsymbol{x})$ as shown in Tab. 7. These ratios stay within reasonable values, in other words, and interestingly, a divergence trained for a corruption $\tau$ may also perform well w.r.t. a different corruption $\tau'$. This shows that a trained BD can generalize across corruptions, and consequently, the associated mirror descent AT also inherits this generalization.

# E   Further results about mirror descent adversarial training

AT is performed using the SGD optimizer for 150 epochs with a learning rate of 0.1 that decays by a factor of 10 each 50 epochs, a batch size of 128, and a weight decay of $5 \cdot 10^{-4}$. These are the same hyperparameters for which RLAT performs the best. The RLAT radius is taken to be 0.08. We also compare against the $L^2$ PGD AT with radius 0.1, which [41] found the most effective for corruption robustness. Fig. 8 shows a sample of the adversarial images found for the contrast and zoom blur corruptions. Fig. 9 shows more adversarial examples for all of 10 BDs (as an extension of Fig. 8).

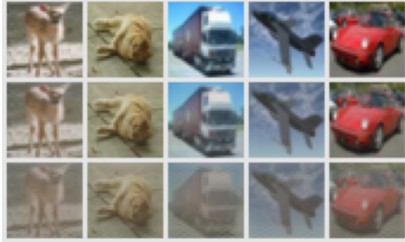

Figure 8: Samples from training images (first row), adversarial examples found by our mirror descent attack using the BD trained for Zoom Blur (second row) and Contrast (third row) corruptions.

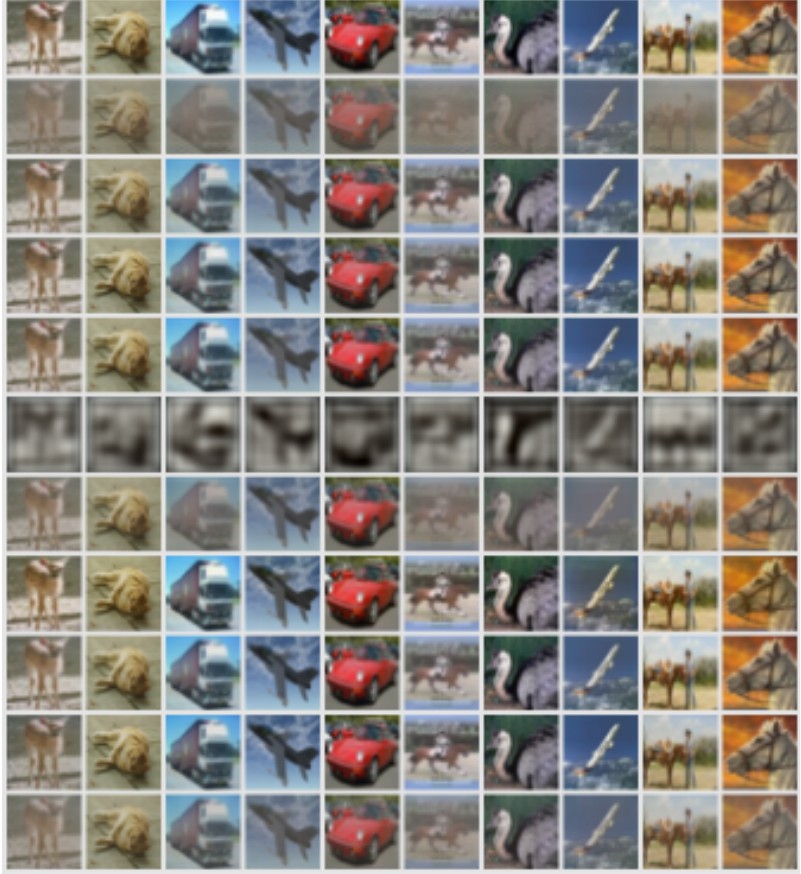

Figure 9: The first row are clean images then each row are adversarial examples found using a BD trained for a different corruption in this order: contrast, zoom blur, fog, brightness, elastic, spatter, jpeg, snow, motion blur, and saturate.

