# OpenReview forum: "Learning Bregman Divergences with Application to Robustness"
_NeurIPS.cc/2024/Conference — NeurIPS 2024 poster_

### Official Review · Reviewer_zPfU · 2024-06-18

**Soundness:** 3
**Presentation:** 3
**Contribution:** 3
**Rating:** 7
**Confidence:** 4

**Summary:**

This paper proposes to use input-convex neural networks to learn Bregman divergences as a means to distinguish semantically meaningful image corruptions from random noise perturbations. The approach is linked to classifier robustness by showing how the associated mirror descent algorithm can be used to perform adversarial training against image corruptions coming from a Bregman ball. Experiments on benchmark corruption datasets show that the proposed method outperforms prior learned similarity metrics in distinguishing corruption from noise and in adversarial training. The proposed method is also shown to generalize quite well to corruptions that it is not trained on.

**Strengths:**

The paper is very well-written, easy to follow, and has great visualizations. The proposed method appears novel and performative, and is certainly of interest to the ML and robustness communities.

**Weaknesses:**

See questions below.

**Questions:**

1. Line 42: "...that has to be convex and with invertible gradient." Do you mean strictly convex?
2. Line 37: The title of this paragraph is "Bregman divergence and mirror descent," yet mirror descent is not discussed at all. Please give a brief description of mirror descent here.
3. Line 64: Please put the footnote number "1" after the punctuation (period). With the footnote number before the punctuation, it makes it look like an exponent. Please also do the same for all other footnotes.
4. Line 73: In (3), do you mean "argmin" instead of "min"?
5. Table 1: Please write out "ICNN" completely as "input convex neural network" here, or define the acronym somewhere in the text before Table 1 for readers unfamiliar with ICNNs.
6. Line 79: This sentence looks strange starting with $\mathbb{B}_h$; I suggest adding "The ball" to the beginning of the sentence.
7. Table 1: Do you mean "strictly convex" instead of "strongly convex" in the text underneath the base function?
8. Line 79: "...but not necessarily convex." This is not true; Bregman balls, as you've defined them, are indeed always convex. Since $h$ is convex, its domain $\mathcal{X}$ is convex, and therefore convexity of the Bregman ball (4) follows from convexity of $\mathcal{X}$ together with convexity of $D_h$ in its first argument. On the other hand, if you were to have defined the ball with respect to a fixed point in the first argument of $D_h$ and varying second arguments, then the set is not necessarily convex.
9. Line 102: I don't recall ever seeing an ICNN defined in terms of Hadamard squares of the input feedthroughs. Can you explain why you are choosing to define your ICNN model using these Hadamard squares, and how the properties of this model might differ from just using the standard linear feedthroughs with arbitrary (not necessarily nonnegative) weights? Your model seems somewhat restrictive in how much influence the feedthrough may have, as its contributions to each preactivation vector are always nonnegative.
10. Line 123: Again, starting a sentence with math looks strange.
11. Figure 4: How come Moco appears to perform on par with or better than the other two prior learning-based methods in Figure 4b, but it's accuracy is shown as 0 across all noise levels in Figure 4a?

**Limitations:**

N/A.

---

> ### Author Rebuttal · Authors · 2024-08-07
>
> We thank the reviewer for the thorough review and for finding our work "novel and performative" and "of interest to the ML and robustness communities". The questions **Q2**, **Q3**, **Q4**, **Q5**, **Q6** and **Q10** are directly incorporated for the next revision. We answer the rest of the questions below:
>
> **Q1 and Q7:** In both cases, we mean strongly convex. This will corrected in the next revision.
>
> **Q8:** Bregman balls are not necessary convex [21]. For example, the Itakura-Saito ball (a Bregman ball for $h(x)= − \log x$) is not convex (Nock et al. 2005).
>
> **Q9**: The definition in line 102 is indeed a special case of an ICNN. This can be seen by expanding equation (5) to:
>
> $u^1 = q^0\left[W^0x + 0 \right]$
>
> $z^1 =  g^0\left[U^1u^1 + V^0x +  b^0 \right]$
>
> $u^l = q^{l-1}\left[W^{l-1}x + 0 \right]$
>
> $z^{l} = g^{l-1} \left[U^lu^{l} + V^{l-1}z^{l-1} + b^{l-1} \right]$
>
> here [1, Proposition 1] imposes that the activation $q^l$ and $g^l$ are convex and non-decreasing and the weights $V^1, .., V^{L−1}$ and $U^1, .., U^{L−1}$ are non-negative. In particular, we can choose the activations $q^l$ to be the Hadamard power $()^{\circ 2}$ and the weights $U^1, .., U^{L−1}$ to be the identity matrix to obtain exactly the equation (5).
>
> So the non-negativity of the weights is a requirement for convexity that we can not avoid here. The Hadamard square is a choice for an activation function that has practical benefits. As we intend to compute the derivative of this network with respect to the input (to obtain $\Psi$), the derivative of the Hadamard square will be linear feedthroughs. We have tried to define the architecture without these Hadamard squares and the results were not satisfactory. This can be appended as an ablation study for the next revision.
>
> **Q11**: The (positive) ratios of Figure. 4b are an average across all 10000 points. This ratio can be very large (>> 1) for few outliers and just below 1 for the other points. This makes the average of the ratios just above 1 (1.05 for d=1 in Figure 4b) but when we measure the accuracy (number of ratios strictly above 1) we get a negligible number of outliers and thus the accuracy will be close to 0.
>
> References:
>
> (Nock et al. 2005) Fitting the smallest enclosing Bregman ball. European Conference on Machine Learning, 2005.

---

> > ### Comment · Reviewer_zPfU · 2024-08-12
> >
> > Thank you for your responses and revisions. I maintain my original score. See more discussion below that I hope will help further improve your manuscript.
> >
> > **Q1 and Q7**: Do you need to assume strong convexity? It is more standard to define Bregman divergences with respect to strictly convex functions, not strongly convex functions (see even your listed reference, Nock et al. 2005).
> >
> > **Q8**: The Bregman ball, as you defined it in (4), is always convex. As you stated on line 72 of your manuscript, the Bregman divergence $D_h$ is convex in its first argument. Therefore, the set $\mathbb{B}_h(\mathbf{x},\epsilon)$ defined in (4) is a convex set, since it is the $\epsilon$-sublevel set of the convex function $\mathbf{x}' \mapsto D_h(\mathbf{x}',\mathbf{x})$. In your reference Nock et al. 2005, they mention that there are two ways of defining balls from Bregman divergences (equations (3) and (4) in their paper), and they mention that the second way in their equation (4), which coincides with your definition of Bregman ball, leads to a convex set, whereas the first way in their equation (3), does not necessarily lead to convexity. Their example of the nonconvex Itakura-Saito ball is an instantiation of their equation (3), which is different from your definition of Bregman ball (and, correspondingly, their equation (4)).
> >
> > **Q9**: "here [1, Proposition 1] imposes that the activation and are convex and non-decreasing" ... "we can choose the activations $q^l$ to be the Hadamard power". Technically, [1,Proposition 1] doesn't apply to the expansion of your proposed architecture, since $q^l(\cdot) = (\cdot)^{\circ 2}$ is not nondecreasing (it is decreasing in a given element when that element ranges over negative values). However, it is pretty obvious that your proposed architecture is input-output convex. My original comment was moreso on your choice of using Hadamard square activation functions for the feedthrough terms. In particular, I suggest clearly asserting in your manuscript that your (5) is a particular example of an ICNN where you choose to impose Hadamard square activations on every feedthrough term, in order to clarify to the reader that not all ICNNs can be written in your form. Also, it would be good to discuss why you chose such a parameterization, over the more general ICNNs used in other works.
> >
> > **Q11**: I see, thanks for the clarification.

---

> > > ### Author Response · Authors · 2024-08-13
> > >
> > > We thank the reviewer for engaging in the rebuttal and for the corrections.
> > >
> > > **Q1 and Q7:** Indeed, strict convexity alone is enough to define the divergence but since our ICNNs is $\alpha$-strongly convex (needed later for the conjugate), one can assume strongly convexity from the beginning.
> > >
> > >
> > > **Q8:** We agree. The Bregman ball as we defined in Eq.4 is what is called the dual Bregman ball which is convex. This convexity explains the success of our projection procedure.
> > >
> > >
> > > > ... since $q^l(\cdot) = (\cdot)^{\circ 2}$ is not nondecreasing (it is decreasing in a given element when that element ranges over negative values). However, it is pretty obvious that your proposed architecture is input-output convex.
> > >
> > > Indeed, it is always the case that the input range is non-negative due to the CELU activations.
> > >
> > >  >  My original comment was moreso on your choice of using Hadamard square activation functions for the feedthrough terms. In particular, I suggest clearly asserting in your manuscript that your (5) is a particular example of an ICNN where you choose to impose Hadamard square activations on every feedthrough term, in order to clarify to the reader that not all ICNNs can be written in your form. Also, it would be good to discuss why you chose such a parameterization, over the more general ICNNs used in other works.
> > >
> > > We will feature this architectural choice more prominently in the next revision along with an ablation study in the appendix.

---

### Official Review · Reviewer_AVtS · 2024-07-08

**Soundness:** 2
**Presentation:** 3
**Contribution:** 1
**Rating:** 4
**Confidence:** 4

**Summary:**

The authors present an approach to learn Bregman divergences that capture perceptual image similarities according to a given dataset.
Relying on two input-convex neural networks, they present a procedure that mimics mirror descent over the learned Bregman divergence.
The procedure is used to learn networks that are robust to image corruptions. Results on CIFAR-10-C subsets are presented.

**Strengths:**

The idea to (attempt to) do mirror descent over learned Bregman divergences in order to train networks robust to corruptions is novel and interesting.

**Weaknesses:**

While the idea in itself is interesting, I do not find neither the technical presentation, nor the provided results convincing.

Most of the motivation of the work derives from the use of Bergman divergences, which come with an associated mirror descent. However, in practice, I do not think the authors can be claiming to do mirror descent, because of the approximation of the inverse map, and because of the lack of a projection operator. Given the two above limitations, I do not think what the authors do would have convergence guarantees even in the convex case. Taking this into account, the stress on the mathematical motivation of the approach seems to be a bit fragile. Sometimes mathematical concepts are introduced without a clear purpose (for instance, the Legendre type, which is then not really necessary to justify their approximation of the inverse map). I would urge the authors to tone down these claims and refrain from saying they are doing mirror descent: I'd rather call it an approach "inspired by mirror descent".

Furthermore, the work assumes that a dataset describing the corruptions is available to learn the divergence: is this a reasonable assumption for datasets such as CIFAR-10-C, which are designed as benchmarks for OOD generalization?

Concerning the results: I do not think the proposed comparisons are fair. Both l2 PGD and RLAT use a threat model which is general and not targeted at specific perturbations. As such, they attain good performance over the entirety of the CIFAR-10-C corruptions. The authors, instead, focus on a small set of perturbations and show results for an algorithm that is explicitly aware of the perturbations the network need to be robust against.

**Questions:**

1) How is the dataset for the training of the Bergman divergence obtained? Is this a holdout from the original CIFAR-10-C?
2) Could the authors add the performance against noise-like perturbations in the comparison against PGD and RLAT?
3) Would the proposed approach scale to ImageNet-C? I do not think scaling to ImageNet-C is necessary, but I think such discussions should be included.
4) The employed ICNN is different from the original work [1] as it displays quadratic terms. I understand that the squared term on x added on top of $z^l$ is useful for strong convexity. What about the quadratic terms in equation (5)?

**Limitations:**

The limitations paragraph is satisfactory.

---

> ### Author Rebuttal · Authors · 2024-08-07
>
> The goal is to generate Bregman divergences from learned base functions $\phi$, that are parametrized neural networks. As a result, the gradients of the base functions $\Psi$ and their inverses $\Psi^{-1}$ are also learned and thus approximated by definition. We see your point on calling it mirror descent and will adapt the wording, but, after doing so, we think it is fair to use the name as it closely follows the general structure and despite the absence of convergence guarantees that, to the best of our knowledge, have never been established even for PGD when adopted for adversarial training by the seminal work of [52]. Nevertheless, we believe that the (meanwhile more, see global comment) extensive experimental results show the effectiveness of our method. Finally, we note that our heuristic is mathematically still a projection (idempotent mapping from a set to a subset).
>
>
> > Furthermore, the work assumes that a dataset describing the corruptions is available to learn the divergence: is this a reasonable assumption for datasets such as CIFAR-10-C, which are designed as benchmarks for OOD generalization?
>
> > I do not think the proposed comparisons are fair. Both l2 PGD and RLAT use a threat model which is general and not targeted at specific perturbations ... be robust against.
>
> To address these two OOD-related concerns, we trained a corruption oblivious Bregman divergence on a different dataset (Berkeley-Adobe Perceptual Patch Similarity) and performed the associated adversarial training on CIFAR-10. The results on CIFAR-10-C outperforms RLAT and PGD especially for the Fog and the Contrast corruptions where PGD and RLAT are known to fail. Please refer to the global rebuttal for details.
>
>
>
> > How is the dataset for the training of the Bergman divergence obtained? Is this a holdout from the original CIFAR-10-C?
>
> The Bregman divergence training algorithm (Sec. 3.2 and Algorithm 1) takes a corruption $\tau$ as an input ($\tau$ is a function that corrupts a given input image). Training is then done on the CIFAR-10 test set, creating pairs, clean and corrupted by $ \tau$. CIFAR-10-C corresponds to the CIFAR-10 test set.
>
> Alternatively, it is possible to dircetly train with (clean, corrupted) pairs as we have done in the new experiments on the Berkeley-Adobe Perceptual Patch Similarity dataset (see global rebuttal for details).
>
>
>
> > Could the authors add the performance against noise-like perturbations in the comparison against PGD and RLAT?
>
> The accuracy for noise categories such as Gaussian is low (52.41\%) which a direct consequence of training algorithm: The core idea is to train the Bregman divergence to consider those noisy images far and thus excluded from the Bregman ball and finally not covered during adversarial training.
>
>
>
> > Would the proposed approach scale to ImageNet-C? I do not think scaling to ImageNet-C is necessary, but I think such discussions should be included.
>
> The Bregman divergence (despite being trained on CIFAR-10 32x32 images) performs well for 256x256 ImageNet images as shown in the rebuttal PDF. Unfortunately, the adversarial training on ImageNet to evaluate on ImageNet-C is beyond the computational resources we have access to (one V100 GPU).
>
>
>
> > The employed ICNN is different from the original work [1] as it displays quadratic terms. I understand that the squared term on x added on top of is useful for strong convexity. What about the quadratic terms in equation (5)?
>
> The equation (5) may look a bit different the equations in [1] but it is a special case that conforms to the conditions of the Proposition 1 in [1]. The equation (5) can be expanded to:
>
> $u^1 = q^0\left[W^0x + 0 \right]$
>
> $z^1 =  g^0\left[U^1u^1 + V^0x +  b^0 \right]$
>
> $u^l = q^{l-1}\left[W^{l-1}x + 0 \right]$
>
> $z^{l} = g^{l-1} \left[U^lu^{l} + V^{l-1}z^{l-1} + b^{l-1} \right]$
>
> here [1, Proposition 1] imposes that the activation $q^l$ and $g^l$  are convex and non-decreasing and the weights $V^1, .., V^{L−1}$ and $U^1, .., U^{L−1}$ are non-negative. In particular, we can choose the activations $q^l$ to be the Hadamard power $()^{\circ 2}$ and the weights $U^1, .., U^{L−1}$ to be the identity matrix to obtain exactly the equation (5).

---

> > ### Comment · Reviewer_AVtS · 2024-08-11
> >
> > I thank the authors for their response. I am happy to see the new experiments showing results on CIFAR-10-C with the divergence learned from BAPPS. However, the corruptions in the comparison table (fog, contrast, zoom blur) appear to be even more cherry-picked in this context where the divergence is learned from another dataset. Given that the baselines (PGD and RLAT) typically improve OOD generalisation over a wide range of corruptions, the comparison still feels quite unfair, casting doubts over the overall empirical effectiveness of the proposed approach. In this sense, the marked decrease in the accuracy to noise-based corruptions should be concerning for an approach presented as a tool to increase general robustness.
> > Finally, the authors state that their addition of the quadratic terms to ICNNs was empirically useful. This detail, along with a comprehensive explanation, should have been prominently featured in the submission.

---

> > > ### Author Response · Authors · 2024-08-12
> > >
> > > We are happy that the new experiments (corruption-oblivious Bregman divergence and the associated Mirror Descent adversarial training) addressed the OOD concerns raised earlier and here we answer the new concerns.
> > >
> > > Before, we want to emphasize that we consider the learning of a meaningful, consistent, applicable Bregman divergence (as confirmed by the experiments and the prototypical and successful application to robustness) a main contribution of this paper and we would imagine that the community could use our pipeline in various settings where non-Euclidean similarity is useful. The new results suggested by you and another reviewer further strengthen our work.
> > >
> > > > Given that the baselines (PGD and RLAT) typically improve OOD generalisation over a wide range of corruptions ...
> > >
> > > We are particularly focusing here to present the results on fog and contrast corruptions only because these corruptions are problematic for both PGD and RLAT:
> > >
> > > From (Kireev et al., 2022):
> > > > Interestingly, for the fog and contrast corruptions, the performance degrades for all methods (see Table 10 in App. H), consistently with the observation made in Ford et al. (2019).
> > >
> > > From (Ford et al., 2019):
> > > > Interestingly, both methods performed much worse than the clean model on the fog and contrast corruptions. For example, the adversarially trained model was 55% accurate on the most severe contrast corruption compared to 85% for the clean model.
> > >
> > > We have shown that our method (trained on a different dataset and oblivious to the corruptions) excels particularly for these problematic corruptions, the improvement in accuracy is about 27%. Also, our method achieves comparable performance on other non-problematic corruptions such zoom blur. This is a notable strength of our method.
> > >
> > > > In this sense, the marked decrease in the accuracy to noise-based corruptions should be concerning for an approach presented as a tool to increase general robustness.
> > >
> > > We respectfully disagree. Robustness against noise is, in this case, neither very important $i)$ nor expected $ii)$:
> > >
> > > * $i)$ Robustness against real-world corruptions such as contrast is more important than robustness against Gaussian noise (that is equivalent to robustness on $L^2$ balls), e.g.:
> > >
> > > > Goodfellow, Shlens, and Szegedy intended $l_p$ adversarial examples to be a toy problem where evaluation would be easy, with the hope that the solution to this toy problem would generalize to other problems. (Gilmer et al., 2018)
> > >
> > > * $ii)$ We train the Bregman divergence by forcing it to consider noisy images far from the original (divergence values for noisy images are in 10000 order of magnitude while divergence to corrupted images are typically below 100). So when we pick a Bregman ball radius (of 100) for adversarial training, the noisy images (gaussian noise) will be automatically excluded. Thus, model trained to be robust in these Bregman balls are not even expected to be robust on noisy images.
> > >
> > > > Finally, the authors state that their addition of the quadratic terms to ICNNs was empirically useful. This detail, along with a comprehensive explanation, should have been prominently featured in the submission.
> > >
> > > These quadratic terms are not a result of random tuning. They are a choice of activation functions obeying the conditions of the ICNN. In fact, the quadratic function is the simplest choice that obeys them (apart from the identity). We will make this clearer in the next revision and add an ablation study in the appendix.

---

### Official Review · Reviewer_DULG · 2024-07-10

**Soundness:** 4
**Presentation:** 4
**Contribution:** 2
**Rating:** 5
**Confidence:** 4

**Summary:**

The authors propose a new method to learn Bregman divergences from raw, high-dimensional data. This method measures similarity between images in pixel space, and considers two images as similar even if one image is corrupted by real-world corruptions, such as blur, changes in contrast, or weather conditions such as fog. The method does this in-part by simultaneously considering real-world corruptions as close to the original image, while noisy perturbations as far from the original image, even when the $L^p$ distance considers noisy perturbations as close. The authors then define adversarial attacks by replacing the projected gradient descent with mirror descent using the learned Bregman divergence. Through adversarial training on this new learned Bregman divergence, they improve the state-of-the-art in robustness.

**Strengths:**

- The authors clearly explain the pipeline of the algorithm and give great explanations for the choices they made (e.g. using equation (7) to approximate $\nabla \bar{\phi}$.)

- The authors make a good case for using Bregman divergences and learning the metric, and it seems like an interesting direction.

- The algorithm seems well-motivated at each step of the pipeline, and it looks like the authors took care to make sure each step follows theory. Figures 1, 2, and 3 are helpful in explaining the motivation.

**Weaknesses:**

- A big weakness is using only one dataset for comparison. Perhaps the authors could show more experiments on ImageNet-C, and/or the Berkeley-Adobe Perceptual Patch Similarity (BAPPS) that was introduced with one of the methods the authors compared against, LPIPS.
  - On the above note, there are a lot of parts to the algorithm, and it's unclear how hard one has to tune the algorithm to make sure each approximation lines up to get an overall, well-performing model. I would have wanted to see a stress test on the pipeline with larger images.
  - If the authors had comparisons on more datasets and of more difficulty than CIFAR10-C, then I would be inclined to raise the score to an accept.

- The proposed method takes longer to train than other standard adversarial training methods, as mentioned in the appendix.

EDIT: After considering the author responses and reading all the reviewers, I have raised my score from a 4 to a 5.

**Questions:**

- Can we see some examples of noisy images that are considered not semantically the same as the original? Under some noise threshold, it seems reasonable for a human to classify a noisy image as the original label, right?

- In Table 3, when using the proposed method, training for one corruption doesn't necessarily perform the best for that corruption (as acknowledged by the authors in the paper). For example, $\text{MD} \thinspace D_{\phi}^{\text{contrast}}$ does not perform the best on contrast corruptions, but rather $\text{MD} \thinspace D_{\phi}^{\text{zoom-blur}}$ does the best. Do you have an explanation for that? I would have liked to see this explored and explained, as it goes against intuition.

- There are a lot of approximations, like computing the inverse map. How do you expect your algorithm to perform with higher resolution images?

**Limitations:**

The authors appropriately acknowledge limitations where appropriate.

---

> ### Author Rebuttal · Authors · 2024-08-07
>
> We thank the reviewer for acknowledging the "great explanations" of the choices we made (especially the approximation of the conjugate $\nabla \overline \phi$), for finding the use of the Bregman divergences for metric learning "interesting direction", and for acknowledging that each step of the pipeline is "well-motivated". We have performed the suggested experiments, and we answer the other concerns below:
>
> > A big weakness is using only one dataset for comparison. Perhaps the authors could show more experiments on ImageNet-C, and/or the Berkeley-Adobe Perceptual Patch Similarity (BAPPS) that was introduced with one of the methods the authors compared against, LPIPS.
>
> Done, we have done two new experiments on BAPPS and evaluation on 256x256 ImageNet images. Please refer to the global rebuttal.
>
> > How do you expect your algorithm to perform with higher resolution images?
>
> The performance on higher resolution (up to 256x256) matches that presented for lower resolution (32x32) as shown by the new experiments on BAPPS and ImageNet. The Bregman divergence successfully distinguishes corrupted from noisy 256x256 ImageNet images despite being trained on 32x32 CIFAR-10 images. The inference is possible because the convolutional layers are followed by an average pooling to size 1x1 before the final feedforward layers.
>
>
> > there are a lot of parts to the algorithm, and it's unclear how hard one has to tune the algorithm to make sure each approximation lines up to get an overall, well-performing model.
>
> The part that needs a tuning effort is the architecture of the base function $\phi$ which is typical for deep learning pipelines. The other parts are fairly standard such as the widely used Adam optimizer.
>
>
>
> > The proposed method takes longer to train than other standard adversarial training methods, as mentioned in the appendix.
>
> Yes, it requires roughly twice the runtime as the standard AT when implemented in PyTorch and run on a single V100 GPU.
>
>
>
> > Can we see some examples of noisy images that are considered not semantically the same as the original? Under some noise threshold, it seems reasonable for a human to classify a noisy image as the original label, right?
>
> Please refer to the rebuttal PDF. Those are 256x256 ImageNet clean samples in the first column, corrupted versions in the second column, and noisy versions (with different noise thresholds) thereafter. Unlike the $L^2$, the trained Bregman divergence considers the corrupted version closer than the noisy versions (as extensively evaluated in the plot and table in Figure 4 of the paper).
>
>
>
> > In Table 3, when using the proposed method, training for one corruption doesn't necessarily perform the best for that corruption (as acknowledged by the authors in the paper). [...] Do you have an explanation for that? I would have liked to see this explored and explained, as it goes against intuition.
>
> This behavior is investigated and to some degree explained in the paragraph "Cross-corruption generalization" of Sec.7 with further experiments reported in Tab.4. For example, the fact that classification models trained with mirror descent under the divergence $D_\phi^{zoom-blur}$ are performing well on contrast can be traced back to the fact that the divergence $D_\phi^{zoom-blur}$ does consider Contrast corruptions close the original images. Why this is the case is not clear but one reason may be that fog/blur/zoom-blur are somewhat similar corruptions.
>
>
> > There are a lot of approximations, like computing the inverse map
>
> Our approach includes two heuristics:
>
> **Approximate inverse map.** This one we consider strong. The approximation is not only theoretically principled but also empirically precise. It is based on the Fenchel conjugate (Fenchel, 1949) as detailed in Section 4. The resulting approximation has satisfactory empirical precision: the mean square error between an image $\boldsymbol{x}$ and its reconstruction $\overline\Psi(\Psi(\boldsymbol{x}))$ (after undergoing the map $\Psi$ and the inverse map $\overline\Psi$) is less than 0.001 on the test set.
>
>
> **Approximate projection**. Indeed, the exact projection to the closest point with respect to Bregman divergence is known to be an open problem (Dhillon & Tropp, 2008) and our approach would immediately benefit from any progress, exact or heuristic. However, our proposed heuristic is successful:
> * it yields valid results (inside the Bregman ball),
> * it results in images with high semantic similarity (as illustrated in Figures 7 and 8),
> * it is fast enough to be incorporated in training,
> * it provides good results in the down-stream task (an accuracy improvement up to 27.16%),
> * mathematically, it is still a projection (idempotent mapping from a set to a subset).

---

> > ### Comment · Reviewer_DULG · 2024-08-10
> >
> > I thank the authors very much for their hard work in addressing my concerns and in the additional experiments. I just have a couple more questions.
> >
> > - Can you provide more context on the rebuttal PDF, such as procedures used to generate those images, where the images came from, etc.? I'm assuming they came from BAPPS?
> >
> > - In the rebuttal pdf, I see that in the 4th row (image of a parrot's right face), under the proposed metric $D_\phi$ the black-and-white image is much closer to the original image than the image in the 3rd column (slight noise). But, in my opinion, most humans would make the the opposite judgement. How would you address such concerns, as I imagine the metric $D_\phi$ is meant to be more "semantic" than the simple L2 distance?
> >
> > Thank you again in working to address my concerns.

---

> > > ### Author Response · Authors · 2024-08-10
> > >
> > > We thank the reviewer for engaging in the rebuttal. We are happy to answer the new questions:
> > >
> > > > Can you provide more context on the rebuttal PDF, such as procedures used to generate those images, where the images came from, etc.? I'm assuming they came from BAPPS?
> > >
> > > Those are 256x256 images from the ImageNet validation set. The first column is for clean samples. The second column are corrupted versions of clean images. The chosen corruption here is Gray color conversion. Thereafter, we generate noisy images (based on the clean image) according to Equ.8 of the paper. For each image (corrupted or noisy) we report the $L^2$ distance to its clean image along with the Bregman divergence $D_\phi$. This Bregman divergence is trained on 32x32 CIFAR-10 training set for the corruption $\tau$ as Gray color conversion following the scheme detailed in Sec. 3.2 and Algorithm 1.
> > >
> > > > In the rebuttal pdf, I see that in the 4th row (image of a parrot's right face), under the proposed metric $D_\phi$ the black-and-white image is much closer to the original image than the image in the 3rd column (slight noise). But, in my opinion, most humans would make the the opposite judgement. How would you address such concerns, as I imagine the metric $D_\phi$ is meant to be more "semantic" than the simple L2 distance?
> > >
> > > We train the Bregman divergence by forcing it to consider corrupted images (gray images in the second column) close to the clean images, and the noisy images as far. The threshold to generate noisy images is the adjustable hyperparameter, $d>0$ in Equ.6. For this particular experiment the noise threshold $d$ is chosen to be relatively low: randomly sampled from $[10^{-7}, 0.99 ]$ (see paragraph "Training details" of Sec. 6 for more details).
> > >
> > > The noise parameter $d$ corresponding to the image in the fourth row, third column is $d\approx0.4$ (which is the $L^2$ of this noisy image divided by the $L^2$ of the corrupted image in the second column). This value of $d$ falls into what we forced the divergence to consider far from the image ($10^{-7}<d<0.99$) so the divergence is behaving as intended. We have chosen these low values of $d$ because they gave the best result once the divergence is used for Mirror Descent adversarial training.
> > >
> > > Nevertheless, in another use-case, the threshold noise $d$ can be increased (for example above 1), so the divergence will consider even images in the seventh column close to the original image if that is what the user wants.

---

### Author Rebuttal · Authors · 2024-08-07

We thank the reviewers for their comments. We responded to each concern in detail in our individual responses. These discussions and also corrections will be incorporated in the next revision. We have strengthened the results of our work by performing an evaluation on the BAPPS dataset (suggested by **Reviewer DULG**) and further trained a corruption-oblivious Bregman divergence to address the Out-Of-Distribution concerns raised by **Reviewer AVtS**. These experiments are detailed below:


**A. Performance on BAPPS.** We applied Bregman learning to the Berkeley-Adobe Perceptual Patch Similarity (BAPPS) dataset (Zhang et al. 2018). Particularly, we considered the two alternative forced choice (2AFC) test. The dataset is a collection of image triplets (reference, distortion 1, distortion 2) and a human judgment stating which of distortions is similar to the reference (so no classification labels). The images and the distortions are diverse from 6 categories. We train our Bregman divergence to mimic the human judgement, more similar distortion closer to the reference. Since we are not using additional data, we compare against the VGG version of LPIPS that does not use ImageNet pre-training. The training pipeline (loss, optimizers, batch size, ... etc.) is similar to that in (Zhang et al. 2018). We report the accuracy results on the 6 test categories of 2AFC below. Our method outperformed LPIPS in all categories except for Frame Interp:

|     | Traditional | CNN-based   |  Superres | Video Deblur | Colorization | Frame Interp |
| :---        |    :----:   |  :----:   | :----:   |:----:   |:----:   |:----:   |
| LPIPS     | 51.41       | 72.10    | 60.46 | 54.25 | 55.18 | 55.55 |
| Bregman (ours)   | 63.65 | 79.57 | 61.04  | 56.95  | 61.63 | 53.73 |

**B. The corruption-oblivious Bregman divergence.** Inspired by the BAPPS experiments, we extended our robustness pipeline and experiment to be oblivious to the corruptions in CIFAR-10-C. Namely, we trained a Bregman divergence on the entire BAPPS that considers distorted versions close to the original and noisy far, exactly following our scheme described in Sec. 3.2 and Algorithm 1 except that the corrupted image is not generated by a corruption $\tau$ but rather it is one of the distortions from BAPPS. BAPPS considers a few dozens of distortions and the divergence is thus not corruption specific, and oblivious to the corruptions in CIFAR-10-C.

We then re-executed adversarial training on CIFAR-10 with this divergence. The results on CIFAR-10-C show again that our method outperforms RLAT and PGD especially for the Fog and the Contrast corruptions where PGD and RLAT are known to fail (e.g., [25] and [41]):

|     | Clean | Fog |  Contrast | Zoom Blur |
| :---        |    :----:   |  :----:   | :----:   |:----:   |
| PGD [41]          | 93.65 | 77.18 | 63.19 | 86.08 |
| RLAT  [41]        | 93.28 | 77.01 | 62.87 | 85.89 |
| Bregman (ours) | 93.61 | 88.00 | 77.70 | 87.12 |


References:

(Zhang et al. 2018) The unreasonable effectiveness of deep features as a perceptual metric. In Proc. CVPR 2018.

---

### Decision · Program_Chairs · 2024-09-25

**Decision:**

Accept (poster)

**Comment:**

This paper proposes an approach for increasing adversarial robustness against some semantic corruptions, such as contrast, blur and fog, which have remained as difficult problems through adversarial training. To do this, this paper proposes to formulate a learned Bregman Divergence, defined by a (strongly) convex (parametric) function. The approach then relies on learning a divergence that measures semantic perturbations (like the ones above) as close and noisy/Gaussian perturbations as far - even if their Lp norms of these are comparable. With such a divergence, the authors propose to learn a robust model by means adversarial training with mirror descent* for the obtained divergence for specific corruptions models. Their numerical experiments demonstrate that models learned under this framework have state-of-the-art robustness against these semantic perturbations.

This paper received mixed opinions and reviews, stimulating feedback from the authors who also provided significantly extended numerical experiments. The discussions among reviewers continued beyond the author-discussion period. In summary:

**Strengths:**

* The novel formulation of learning robust models for semantic perturbations under a learned Bregman divergence,
* The improved robustness obtained for some corruptions.

**Weaknesses:**

* The algorithm implemented in practice is not mirror descent exactly (which would be an important analytical benefit of employing Bregman divergences), given that two of the necessary steps (invertibility of the mirror map and projection onto a Bregman divergence ball) are hard and are therefore solved approximately.
* The initial numerical experiments were carried out on relatively small dimensions, and there are concerns on whether the obtained models are robust more generally rather than with respect to the chosen semantic corruption models.

**Conclusion:**

Upon reading all discussions and opinions, and reading the paper, I recommend this paper be accepted due to the following reasons:

1) The formulation is innovative and seems to lead to significantly better results, but some of the steps are approximations. Adversarial robustness to semantic perturbations is a hard problem, and the presented approach might provide a general framework to make progress on this front. While there were some issues about the way the claims were phrased by the authors, I trust they will incorporate this feedback into the final revision.

2) The authors have significantly expanded their numerical results, showing that the method can be scaled up to higher dimensional data during the rebuttal phase. Most of the reviewers agreed that the revised experiments are convincing and sufficient to justify acceptance.